# Comparison of species-specific qPCR and metabarcoding methods to detect small pelagic fish distribution from open ocean environmental DNA

Zeshu Yu[1,2], Shin-ichi Ito[1]*, Marty Kwok-Shing Wong[1], Susumu Yoshizawa[1], Jun Inoue[1], Sachihiko Itoh[1], Ryuji Yukami[3], Kazuo Ishikawa[1,2], Chenying Guo[1,4], Minoru Ijichi[1,5], Susumu Hyodo[1]

1 Atmosphere and Ocean Research Institute, The University of Tokyo, Kashiwa, Chiba, Japan, 2 Graduate School of Agricultural and Life Sciences, The University of Tokyo, Bunkyo, Tokyo, Japan, 3 Fisheries Resources Institute, Japan Fisheries Research and Education Agency, Yokohama, Kanagawa, Japan, 4 State Key Laboratory of Tropical Oceanography, South China Sea Institute of Oceanology, Chinese Academy of Sciences, Guangzhou, Guangdong, China, 5 Department of Biological Sciences, Tokyo Metropolitan University, Hachioji, Tokyo, Japan

* goito@aori.u-tokyo.ac.jp

**Data Availability Statement:** All relevant data are within the manuscript and its Supporting Information files.

## Abstract

Environmental DNA (eDNA) is increasingly used to noninvasively monitor aquatic animals in freshwater and coastal areas. However, the use of eDNA in the open ocean (hereafter referred to OceanDNA) is still limited because of the sparse distribution of eDNA in the open ocean. Small pelagic fish have a large biomass and are widely distributed in the open ocean. We tested the performance of two OceanDNA analysis methods—species-specific qPCR (quantitative polymerase chain reaction) and MiFish metabarcoding using universal primers—to determine the distribution of small pelagic fish in the open ocean. We focused on six small pelagic fish species (*Sardinops melanostictus*, *Engraulis japonicus*, *Scomber japonicus*, *Scomber australasicus*, *Trachurus japonicus*, and *Cololabis saira*) and selected the Kuroshio Extension area as a testbed, because distribution of the selected species is known to be influenced by the strong frontal structure. The results from OceanDNA methods were compared to those of net sampling to test for consistency. Then, we compared the detection performance in each target fish between the using of qPCR and MiFish methods. A positive correlation was evident between the qPCR and MiFish detection results. In the ranking of the species detection rates and spatial distribution estimations, comparable similarity was observed between results derived from the qPCR and MiFish methods. In contrast, the detection rate using the qPCR method was always higher than that of the MiFish method. Amplification bias on non-target DNA and low sample DNA quantity seemed to partially result in a lower detection rate for the MiFish method; the reason is still unclear. Considering the ability of MiFish to detect large numbers of species and the quantitative nature of qPCR, the combined usage of the two methods to monitor quantitative distribution of small pelagic fish species with information of fish community structures was recommended.

**Funding:** SI, JP21H04735, The Japan Society for the Promotion of Science (JSPS) KAKENHI, https://www.jsps.go.jp/english/. The funders had no role in study design, data collection and analysis, decision to publish, or preparation of the manuscript. SI & SH, The OceanDNA project, The University of Tokyo Future Society Initiative, https://www.u-tokyo.ac.jp/adm/fsi/ja/projects/sdgs/projects_00103.html. The funders had no role in study design, data collection and analysis, decision to publish, or preparation of the manuscript.

**Competing interests:** The authors have declared that no competing interests exist.

## Introduction

Due to the vastness of the ocean and the difficulty of observation during stormy weather, many aspects of the migration and distribution patterns of marine fish species, including important economic species, remain unclear. Surveying fish distribution by net catching seems to be the most reliable method. However, the escaping ability of fish influences the catch efficiency, and this method requires an enormous amount of time in the open ocean, which makes it difficult to obtain high resolution data.

In recent years, a new technique for investigating fish distribution in water, environmental DNA method, has been developed. Environmental DNA (eDNA) is DNA that creatures shed into their living environment, including water, sediment, soil, and even air [1–4]. Surveys using eDNA have been performed in rivers, lakes, and coastal areas [5–9]. However, the eDNA method faces more challenges in the open ocean [10, 11]. The validity of 'eDNA survey in the open ocean' (hereafter OceanDNA) is unclear, given the lower density of creatures like fish and zooplankton in the open ocean [10, 11]. For example, zooplankton were detected in the coastal region, but not in the open waters, in 1.5 L water samples using COI metabarcoding [11]. The foregoing indicates that while OceanDNA has the potential to be a valuable fish survey method, careful feasibility testing and efficient performance protocols are necessary.

Monitoring aquatic species using eDNA always includes the following sequential steps: (1) collecting the environmental samples (water sampling and filtering in fish species surveys), (2) eDNA extraction, and (3) eDNA analysis [1–3]. eDNA analysis can obtain information that includes such as distribution patterns of specific species or species diversity in a habitat [12, 13]. The two primary eDNA analysis methods are: species-specific polymerase chain reaction (PCR) and eDNA metabarcoding [13]. Species-specific PCR includes conventional PCR (PCR), quantitative PCR (qPCR), and digital PCR (dPCR) [14–16]. Species-specific PCR methods are simpler and quicker (several hours to perform PCR followed by analysis) and are economical when the focus is on a few species [14, 15]. In contrast, eDNA metabarcoding can detect many species in a single analysis, which is advantageous in biodiversity or multispecies resource surveys. eDNA metabarcoding uses primers designed for conserved regions to perform PCR, followed by high-throughput sequencing (HTS) and classification of the eDNA into taxonomic units by blasting with a DNA database [9, 17–21]. For fish species, a new eDNA metabarcoding method called MiFish was developed in 2015. MiFish uses primers that target a hyper-variable region (~160–190 bp) of the fish 12S rRNA gene. This region contains sufficient information for identification to the species or genus levels [19]. MiFish has been widely applied over six continents and the Palmyra Atoll in the eastern equatorial Pacific [19, 21].

However, both qPCR and eDNA metabarcoding face observational biases due to the degradation of eDNA, advection by ocean currents, and the inhibition of DNA amplification by additional substances [22–24]. In addition, amplification bias of high-throughput sequencing (HTS) can influence the sensitivity of eDNA metabarcoding [25–28]. High abundance of eDNA may inhibit the amplification of low-abundance eDNA through the competition of binding with metabarcoding primers [25–27]. Both issues can result in the loss of some species in eDNA metabarcoding analysis.

In a study that identified three invasive mosquito species (*Aedes albopictus*, *Ae. japonicus* and *Ae. koreicus*) from natural freshwater bodies, both eDNA analysis methods showed "reliable and congruent results" [29]. Another study involving the Mediterranean fanworm *Sabella spallanzanii* suggested that species-specific PCR (qPCR and ddPCR) had nearly double the probability of detection compared with eDNA metabarcoding [30]. Some other studies also used both species-specific PCR and eDNA metabarcoding [31, 32]. However, research

comparison of the reliability and efficiency of species-specific PCR and eDNA metabarcoding in the open ocean has been insufficient. Thus, to help determine the proper OceanDNA method for marine fish surveys, we compared species-specific qPCR and eDNA metabarcoding for OceanDNA to determine whether their results are comparable and to develop a feasible and optimal protocol for a new open ocean fish survey.

A series of OceanDNA surveys on 247 seawater samples collected from the Kuroshio Extension was performed using the qPCR and eDNA metabarcoding analysis methods. For eDNA metabarcoding, we used the MiFish pipeline [19, 33]. We focused on six small pelagic fish: Japanese sardine (hereafter sardine, *Sardinops melanostictus*), Japanese anchovy (hereafter anchovy) *Engraulis japonicus*, chub mackerel (*Scomber japonicus*), blue mackerel (*Scomber australasicus*), Japanese jack mackerel (hereafter jack mackerel, *Trachurus japonicus*), and Pacific saury (hereafter saury, *Cololabis saira*). The selection of these species reflected their abundance in the surface layers, their importance as target fishery species, and the recent development of a real-time qPCR method for their detection (primers listed in S2 Table). We believe that the acquired data will aid in the establishment of OceanDNA, as a useful tool for evaluating fish distribution in the open ocean, where the concentration of fish eDNA is expected to be scarce, has been insufficient.

## Materials and methods

### Water sampling

Sea water samples were collected during the KS-18-5 research cruise in the Kuroshio Extension area by the R/V Shinsei-Maru research vessel. OceanDNA seawater samples were collected from 19 water sampling stations. The stations were arranged on two lines (nine stations from B-line and ten stations from C-line; Fig 1 and S1 Table), and from 13 depths (0, 5, 10, 15, 25, 30, 50, 80, 100, 125, 150, 200, and 300 m). In total, 247 seawater samples were collected, including 117 B-line samples and 130 C-line samples. Compared with the sea surface velocity and temperature field, both lines extended from the Kuroshio Extension to the mixed water region in the north.

Water samples were collected from water depths of 5 to 300 m using Niskin bottles combined with a conductivity temperature depth (CTD) system. This system can accurately determine depth. The surface (0 m) water samples were collected using a clean bucket. Each water sample was stored in a fresh plastic bag that had been co-washed three times (Rontainer, Seki-sui Chemical Co., Ltd., Tokyo, Japan). Approximately 7 L was weighed and then immediately filtered using a Sterivex-GP pressure filter unit with a 0.22 μm pore size (Merck Millipore, Burlington, MA, USA). To perform filtering, the inlet end of the Sterivex-GP pressure filter unit was attached to the 1/4 inch HB to M Luer lock (XX3002564; Merck Biopharma Co., Ltd., Tokyo, Japan), which was assembled into one end of a peroxide-cured silicon pump tube (L/S25, 96400–25; Yamato Scientific Co., Ltd., Tokyo, Japan). The pump tube was then fixed by tube cartridge (07519–70; Yamato Scientific Co., Ltd.) to the multi-channel pump head (07519–06; Yamato Scientific Co., Ltd.). The pump head was assembled into a digital pump (07528–10; Yamato Scientific Co., Ltd.). Finally, through the peroxide treatment silicon pump tube, the digital pump (rotation speed set to 60 rpm) pushed the water sample through the Sterivex-GP pressure filter unit from the plastic bag. Before we collected water samples, all silicon tubes, tube connectors, and stoppers were treated with 1% bleach (sodium hypochlorite solution) and cleaned with Milli-Q water to prevent contamination before filtering. After filtering, the plastic bag with a residual water sample was weighed, and the weight difference before and after filtering was used as the filtered sea water mass. We connected a 50 mL disposable syringe to the Sterivex filter unit through a disposable connector (discarded every time

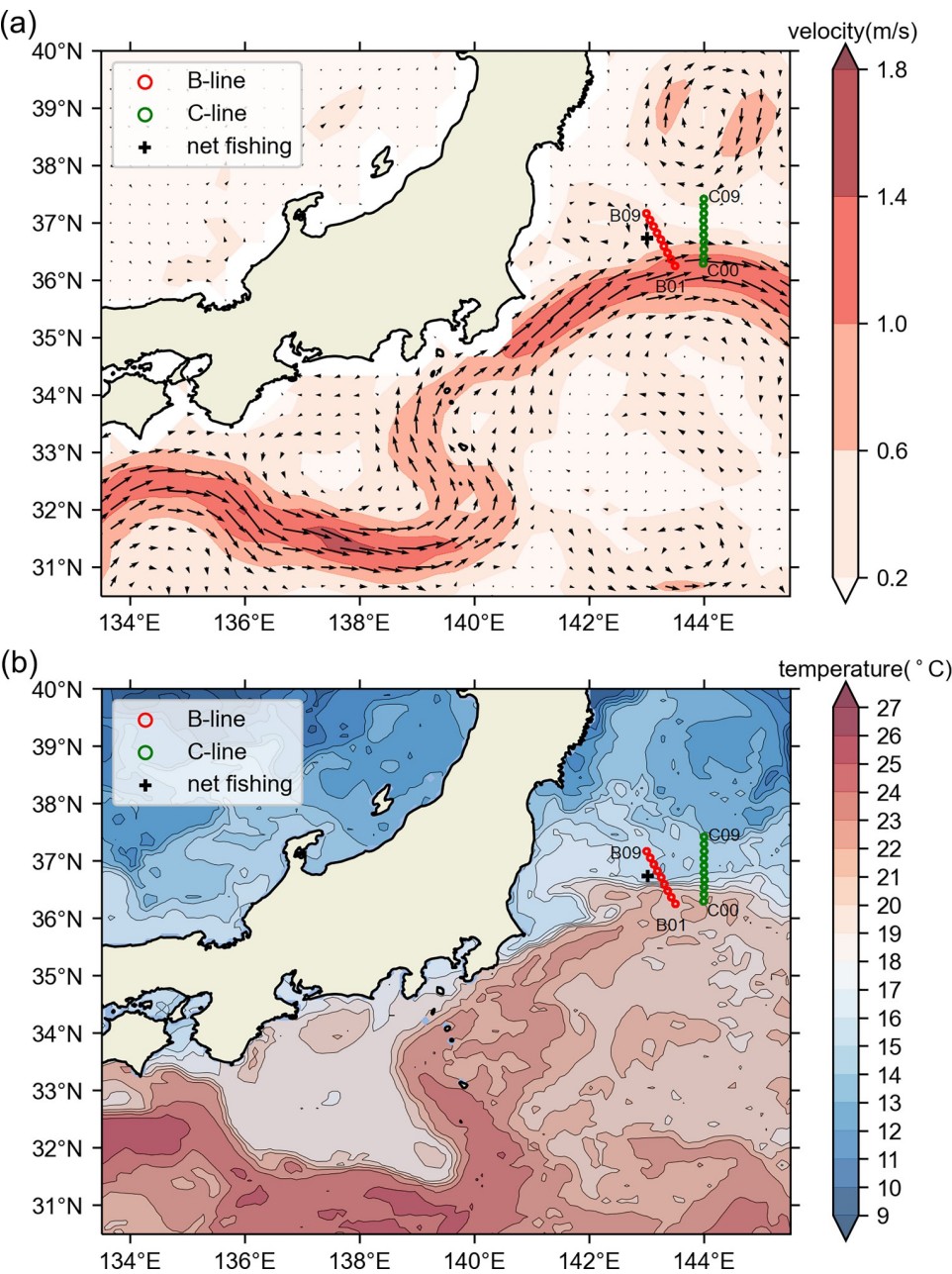

**Fig 1. Positions of OceanDNA sampling stations in the Kuroshio Extension area.** The red circles represent stations B01–B09, the green circles represent stations C00–C09, and the plus mark represents the net sampling location. Maps were made in Spyder (Python 3.7) using Natural Earth data. Free vector and raster map data @ naturalearthdata.com. (a) Colors represent current speeds and arrows represent current velocity vectors on the sea surface (5-day mean from May 10 to 14, 2018) from Ocean Surface Current Analysis Real-time data [34] (https://podaac.jpl.nasa.gov/Data hosted and openly shared by the PO.DAAC, without restriction, in accordance with NASA's Earth Science program Data and Information Policy). (b) The contours and colors represent sea surface temperature (˚C) distribution on May 10, 2018, from GHRSST Level 4 OSTIA Global Foundation Sea Surface Temperature Analysis data [35] (hosted and openly shared by the PO.DAAC, without restriction, in accordance with NASA's Earth Science program Data and Information Policy).

after dealing with a sample) and then excluded the resident water in the Sterivex filter unit by air pressure applied manually. After excluding the sea water completely from the Sterivex filter unit, we used a 5 mL disposable syringe (discarded every time after dealing with a sample) to add 2.0 mL RNAlater Stabilization Solution (Thermo Fisher Scientific, Waltham, MA, USA) into each Sterivex filter unit to immerse the whole membrane, which was then kept for 12 h at 4 ˚C. Then, Sterivex filter units were immediately stored at -30 ˚C until DNA extraction.

## The eDNA extraction and purification

The extraction and purification of eDNA from Sterivex were performed by Bioengineering Lab. Co., Ltd. (Kanagawa, Japan). The procedure was based on a previously described protocol [19]. Before starting the extraction, silicon tubes, tube connectors, and stoppers were treated with 1% bleach and cleaned with Milli-Q water to prevent contamination. Sterivex filter units were kept on ice until the RNAlater was thawed. DNA extraction was performed using Charge Switch Forensic DNA Purification Kit (Thermo Fisher Scientific). Briefly, after removing RNAlater, 2 mL lysis mix (containing Lysis Buffer and 20 μL Proteinase K) was added to each Sterivex filter unit. The filter units were incubated at 55 ˚C for 30 min. Supernatant from each filter unit was collected in a new microcentrifuge tube. ChargeSwitch® Magnetic Beads (Thermo Fisher Scientific) were added to bind to the DNA. After washing the magnetic beads with Wash Buffer, DNA combined with Magnetic Beads was eluted with 150 μL of Elution Buffer for each sample and then transferred to a new microcentrifuge tube. Finally, the eluted DNA solutions were purified by adding an equal volume of AMPure XP Reagent (Beckman Coulter Inc., Brea, CA, USA).

## Metabarcoding of fish (MiFish)

MiFish primers were used to perform eDNA metabarcoding as previously described [19]. The primers and reaction settings are shown in the S1 Fig. The details of the MiFish method are provided in the S1 Text. The MiFish procedure was performed on all 247 samples. The library preparation of the MiFish method failed in 63 samples (49 B-line samples and 14 C-line samples). In the remaining 184 samples, the average number of raw and clean reads was 600,718 and 138,476, respectively. The details of the MiFish results are shown in the S3–S5 Tables.

## The qPCR assay

Real-time qPCR assays were performed on the sardine, anchovy, chub mackerel, blue mackerel, jack mackerel, and saury [36]. Six species-specific probes and qPCR primers were designed in D-loop region of mitochondrial DNA. The detailed primer sequence information is listed in S2 Table. Each 10 μL qPCR reaction system contained 5 μL of 2× TaKaRa Probe qPCR Mix (TaKaRa Bio Corporation, Kusatsu, Japan), 900 nM forward primers and 900 nM reverse primers, 250 nM fluorescent probe, and 2.5 μL of sample/standard eDNA. We performed qPCR using an ABI 7900HT real-time PCR system (Applied Biosystems, Foster City, CA, USA). The thermal cycling profile comprised 95 ˚C for 10 min and 50 cycles of 95 ˚C for 15 s, and 65 ˚C for 1 min. The qPCR standards were prepared by diluting the plasmids containing the specific sequences at $10^1$–$10^7$ copies per 2.5 μL in each reaction. Each sample was assayed in duplicate, and the detailed qPCR results are shown in S6 Table.

## Data analyses

Presence detection indicated that the target fish quantity was higher than 0 in any of the duplicate assays for the qPCR method, or the ratio of the target operational taxonomic units

(OTUs) in the total OTU read was >1% in a sample using the MiFish method. The latter 1% criterion for the MiFish method was used to avoid false presence [37, 38]. In this study, species-specific qPCR was performed on the aforementioned six pelagic fish. However, because of the similarity in DNA sequences, OTU identification in MiFish could not clearly distinguish the DNA sequences of chub mackerel and blue mackerel. Thus, we used "chub/blue mackerel" to represent either chub mackerel or blue mackerel and treated "chub/blue mackerel" as the target fish. As a result, the comparison of eDNA metabarcoding and qPCR was performed on five targets: chub/blue mackerel (chub mackerel and blue mackerel), sardine, anchovy, jack mackerel, and saury. The MiFish procedure could not be applied to 63 OceanDNA samples in which the library preparation failed (S3 Table). Therefore, we excluded their data before comparing the performance of the qPCR and MiFish methods.

To examine the correspondence of the survey results of the species-specific qPCR and MiFish methods, Pearson's Phi-coefficients were calculated based on presence/absence detection data to assess the correlation between qPCR (presence = 1/absence = 0) and MiFish (presence = 1/absence = 0). The value of the Pearson's Phi-coefficient ranges from -1 to +1, where +1 (-1) indicates perfect agreement (disagreement) and 0 indicates no relationship. As the precise meaning of the coefficients (strengths of relationships) depends on the freedom of data, we also calculated the p-value of Fisher's exact test.

To compare the presence data distributions between the qPCR and MiFish methods, spatial distribution figures were made using matplotlib with Spyder (Python 3.7) based on presence/absence data of the qPCR and MiFish methods, DNA quantity data of the qPCR method, space data (depth, longitude, latitude), and environmental data (temperature and chlorophyll-a concentration, showed in S9 Table). The data were recorded by the CTD system during the seawater sampling.

As shown in the Results section, the detection rate of the qPCR method was higher than that of the MiFish method, whereas the presence distributions of both methods were similar. To find the possible reasons for the different detection rate of the qPCR and MiFish methods, we tested two hypotheses. One hypothesis posited that the detection limit of the qPCR method is lower than that of the MiFish method. The other hypothesis posited that OceanDNA amplification of target species is inhibited by OceanDNA amplification of non-target species (amplification bias) in the MiFish method. To test whether the two methods have different DNA quantity detection limits, we divided the samples in which qPCR detected the target species into different groups according to the OceanDNA quantity of target species measured by qPCR and then compared the MiFish detection loss rate among all groups. The MiFish detection loss rate was defined as the percentage of samples in which MiFish failed to detect target species. To test the influence of the amplification bias in the MiFish method, we divided the samples in which qPCR detected the target species into the group in which target fish was not detected by MiFish (MiFish-0/qPCR-1 samples) and the group in which MiFish also detected the target fish (MiFish-1/qPCR-1 samples). We then compared the number of non-target OTUs and the chlorophyll-a concentration between these two groups using the Mann-Whitney U test to test the statistical significance. Non-target OTUs were OTUs that belonged to the non-target fish. Chlorophyll-a concentration was included for comparison because it represents the algae quantity, and algae DNA could not be amplified using the MiFish method.

## Results

### Comparison between results from OceanDNA and from net survey

As our purpose of developing OceanDNA methods is to survey fish distribution in the open ocean, we firstly compared our OceanDNA data with direct net sampling survey results. The

net sampling data used were obtained by a pelagic trawling covered depth of 0–25 m that were performed during a cruise by the R/V Soyo-Maru vessel of the Japan Fisheries Research and Education Agency (net sampling station is shown in Fig 1), on May 11, 2018. The target fish species caught in the net comprised 158 sardine (40.85–66.80 mm body length, 0.85–3.74 g body weight; 0.246 kg in total), 289 chub/blue mackerel (16.99–64.08 mm fork length. 0.04–2.75 g in body weight; 0.044 kg in total), and 40 anchovies (34.99–68.68 mm body length, 0.35–3.24 g in body weight; 0.044 kg in total). The chub/blue mackerel were small in size and it was difficult to distinguish the two species morphologically. The abundance was highest in sardine, followed by chub/blue mackerel and anchovy. We chose the OceanDNA data from station B06 (sampled on May 11, 2018) for comparison because it is the closest to the net sampling point (Fig 1). Both the qPCR and MiFish methods detected the presence of sardine, chub/blue mackerel, and anchovy at B06, with sardine being most prevalent (Fig 2). The result of OceanDNA method mirrored that of the net sampling.

## Detection rate comparison of target species between MiFish and qPCR

The detection results (presence/absence) from both qPCR and MiFish methods showed that sardine and chub/blue mackerel were the most common among the five target fish in both B-line and C-line ("all detection rate" always >20%; S7 Table). For these two common fish, as well as for anchovy, the detection rate using the qPCR method appeared to always be much higher (more than twice) than that of the MiFish method, in both B-line and C-line samples (Fig 3). In the case of saury, the detection rate using the MiFish method was higher than that of the qPCR method (the detection rates were all <4% using both methods and on both lines). Jack mackerel was almost undetectable in both the B- and C-lines, by qPCR or MiFish methods.

On the contrary, the detection rate ranking of the five target fish was the same on the B-line and C-line in each method (Fig 3). For qPCR data, the detection rate ranking for both lines was sardine > chub/blue mackerel > anchovy > saury > jack mackerel. The detection rate ranking of MiFish data on both lines was sardine > chub/blue mackerel > saury > anchovy > jack mackerel. Although the qPCR and MiFish methods have different detection rates for each target fish, the detection rate rankings from the two methods were comparable.

## Correlation between MiFish and qPCR on target species

Since similarities and differences were observed between the results of the MiFish and qPCR methods, we examined the Phi-coefficients between the detection results from the two methods. Since the relative abundance ranking among the five target fish was the same in the B-line and C-line samples and the two lines were not far from each other (Fig 1), we summed the data of these two lines in the Phi-coefficient analyses. Phi-coefficient examination was performed on presence/absence data (0 = absence, 1 = presence) of each target fish (except jack mackerel, as it was only detected in one sample) from all 184 OceanDNA samples in which the MiFish library preparation was successful (Fig 4 and Table 1), and a positive Phi-correlation was detected in every examination (Table 1). Statistically significant positive correlations were found for chub/blue mackerel, sardine, and saury (chub/blue mackerel: Phi $\approx$ 0.17, $p \approx$ 0.0199; sardine: Phi $\approx$ 0.22, $p \approx$ 0.0017; saury: Phi $\approx$ 0.37, $p \approx$ 0.0380; Fisher's exact test). In addition, statistically significant positive correlations were observed (when not including jack mackerel: Phi $\approx$ 0.33, $p \approx$ 6.6e-19; when including jack mackerel: Phi $\approx$ 0.35, $p \approx$ 1.0e-21) from the pooled data (Table 1). The Phi- coefficient examination also revealed the similarity of detection results between the MiFish and qPCR methods.

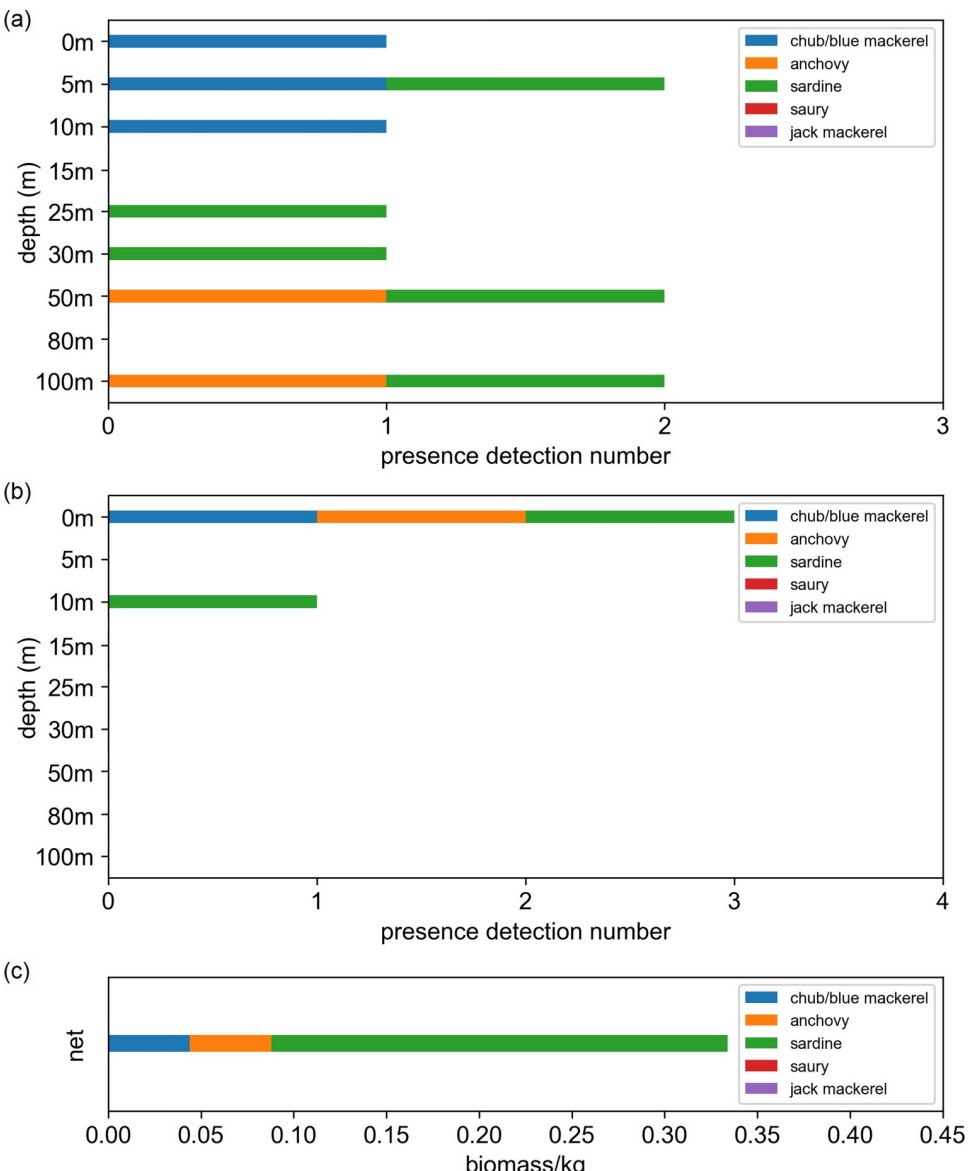

**Fig 2. OceanDNA detection results of five target fish at station B06 and fish caught in net sampling.** Detection results using (a) qPCR method and (b) MiFish method, and (c) fish biomass data from the net sampling (pelagic trawling, at depth of 0–25 m).

## Spatial distribution comparison of target species between MiFish and qPCR

The spatial distribution was analyzed in each target fish using detection data from the qPCR and MiFish methods, as well as the position information of each OceanDNA sample. Because the number of samples detected in anchovy, saury, and jack mackerel were very low (presence detection sample numbers were always <10), only the spatial distribution results of the high-detection-rate chub/blue mackerel and sardine were compared (Figs 5 and 6).

The spatial distribution area of the MiFish presence detection samples was narrower than that from the qPCR method, as expected based on the lower detection rate of the MiFish method (Figs 5 and 6). However, similar distribution patterns were found of the MiFish and

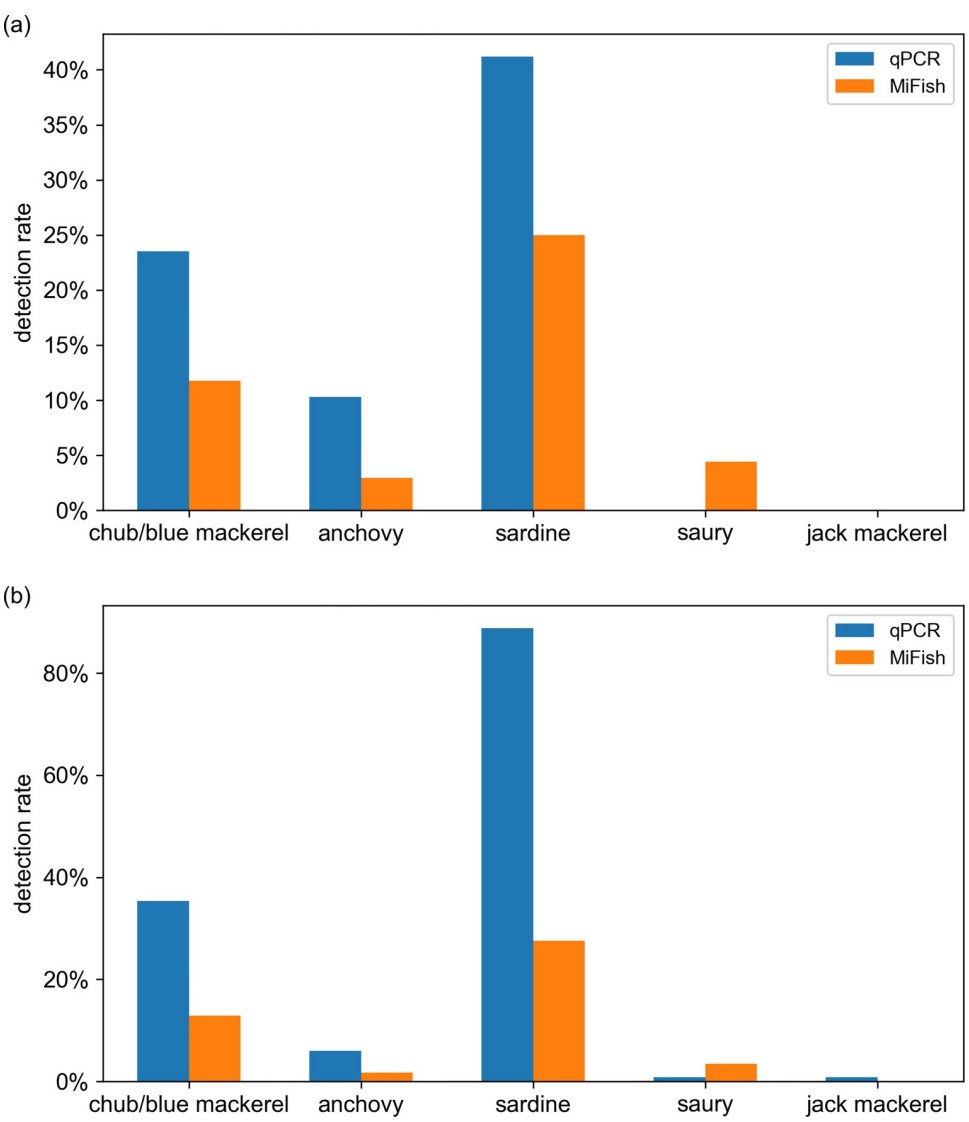

**Fig 3. Detection rate ranking of five target fish based on qPCR data and MiFish data.** Data based on samples collected on the B-line (a) and C-line (b). The 'detection rate' is calculated as: detection number / whole sample number of B-line (or C-line). chub/blue mackerel = *Scomber japonicus* and *Scomber australasicus*, sardine = *Sardinops melanostictus*, anchovy = *Engraulis japonicus*, jack mackerel = *Trachurus japonicus*, saury = *Cololabis saira*.

qPCR methods for both sardine and chub/blue mackerel on the two lines, especially for B-line sardine and C-line chub/blue mackerel. In detail, for B-line collected sardines, the qPCR and MiFish results showed a distribution pattern comprising no sardine distribution at stations 1 and 2, distribution depth of sardine shifted to shallower from stations 4 to 5, and the distribution depth of sardine shifted to deeper from stations 6 to 7 and 8. For C-line chub/blue mackerel, both the qPCR and MiFish results showed similar patterns, with the distribution of chub/blue mackerel around stations 4–6 showing shallower distribution than the others.

## Influences of detection limit and amplification bias on detection rate

As possible sources to generate the detection rate difference between MiFish and qPCR found in this study (Fig 3 and S7 Table), we tested the influence of the detection limit and

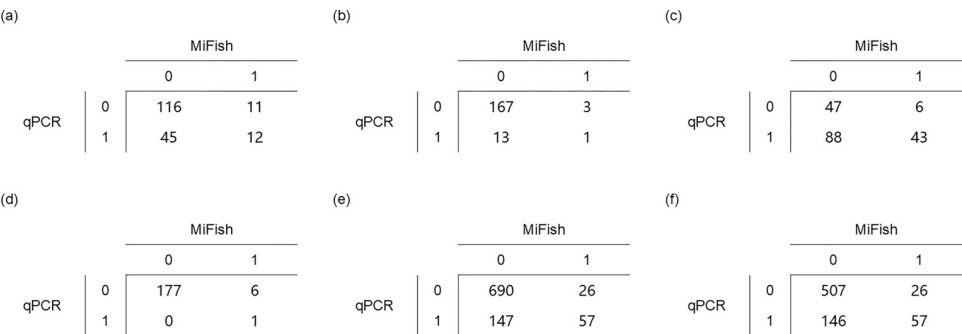

**Fig 4. Comparison tables for Phi-coefficients analysis of detection performance among the five target fish.** (a) chub/blue mackerel (*Scomber japonicus* and *Sc. australasicus*), (b) anchovy (*Engraulis japonicus*), (c) sardine (*Sardinops melanostictus*), (d) saury (*Cololabis saira*), (e) pooling (sum data of all five target fish, chub/blue mackerel, sardine, anchovy, saury and jack mackerel), (f) pooling-4 (sum data of (a) to (d)). 0 = absence, 1 = presence.

amplification bias in MiFish. For the detection limit effect, the dependency of MiFish detection loss rate on OceanDNA quantity of target species measured by qPCR was investigated. The histograms of the MiFish presence/absence detection based on the qPCR determined OceanDNA quantity showed that the highest MiFish detection loss rate (percentage of the samples in which MiFish failed to detect the target species) of chub/blue mackerel and sardine appeared in the bins of 0–0.5 copies/μL and 0.5–1.0 copies/μL OceanDNA quantity (Fig 7). The MiFish detection loss rate decreased markedly for samples in which OceanDNA quantity exceeded 5.0 copies/μL (Fig 7). However, the MiFish detection loss rate was still high (66.7% for chub/blue mackerel and 65.4% for sardine) in the bin of 1.0–5.0 copies/μL OceanDNA quantity (Fig 7).

For amplification bias in MiFish, we investigated the influence of non-target OTUs and chlorophyll-a. Non-target OTUs were those that did not belong to the target fish; these represented fish DNA that was amplified in the library preparation but did not belong to the target fish. In the MiFish-0/qPCR-1 samples, the number of non-target OTUs was higher than that in the MiFish-1/qPCR-1 samples (Table 2). The increase was approximately 35% higher in chub/blue mackerel, 153% in sardine, and 115% in anchovy. All increases were statistically significant ($p < 0.001$, Mann-Whitney U test). Conversely, the chlorophyll-a concentration was lower in the MiFish-0/qPCR-1 samples than in the MiFish-1/qPCR-1 samples for anchovy and sardine (Table 2).

## Discussion

eDNA has received much attention as a means of investigating the distribution of open ocean fish. Metabarcoding and species-specific PCR are the two main eDNA analysis methods. The MiFish method is a metabarcoding method developed for fish that focuses on a part of the fish 12S rRNA gene. Although it is a relatively new method, the MiFish method has been widely

**Table 1. Coefficients (Phi-coefficient value and Fisher's exact test) of detection performance among the five target fish.**

|  | chub/blue mackerel | anchovy | sardine | saury | pooling-4[a] | pooling[b] |
|---|---|---|---|---|---|---|
| Phi-coefficient | 0.17 | 0.10 | 0.22 | 0.37 | 0.33 | 0.35 |
| $p$ | <0.05 | >0.05 | <0.05 | <0.05 | <0.05 | <0.05 |

[a] *pooling-4* is the sum of chub/blue mackerel, anchovy, sardine, and saury

[b] *pooling* refers to the sum of chub/blue mackerel, sardine, anchovy, saury, and jack mackerel.

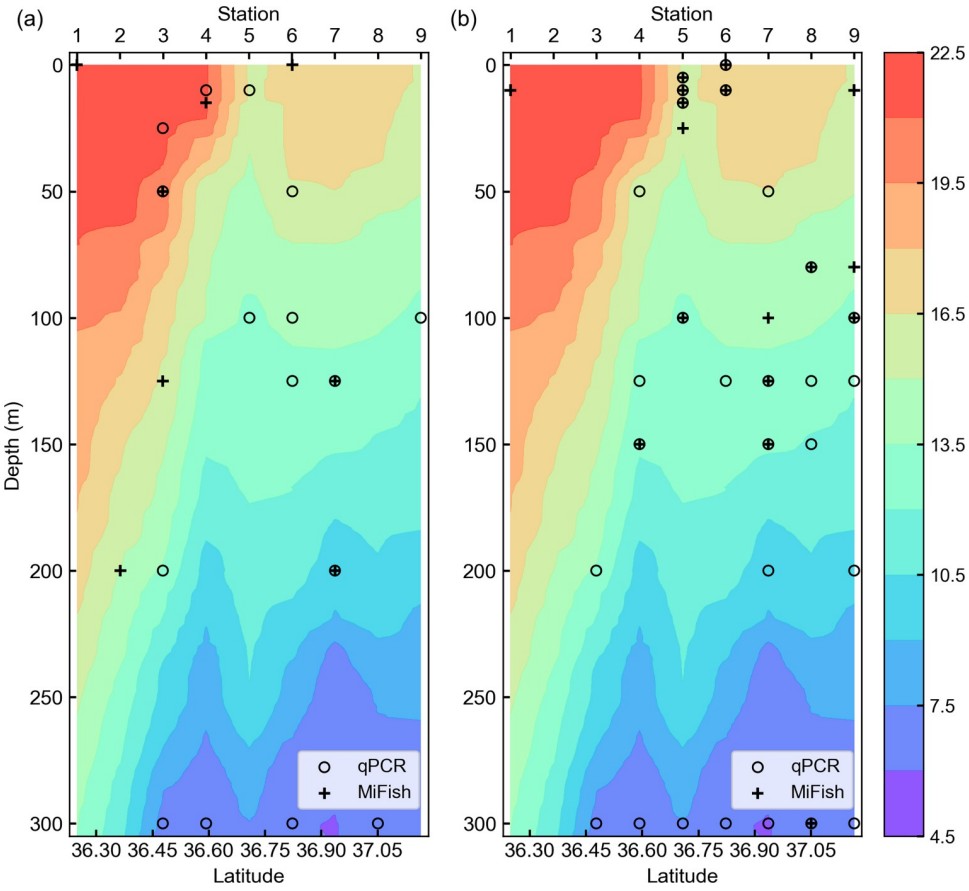

**Fig 5. Comparison of spatial distribution pattern detected by qPCR and MiFish methods in the B-line vertical section.** (a) Spatial distribution of chub/blue mackerel (*Scomber japonicus* and *Sc. australasicus*). (b) Spatial distribution of sardine (*Sardinops melanostictus*). Plus marks represent the presence detection samples found by the MiFish method and open circles represent the presence detection samples found by the qPCR method. Background colors represent water temperature (˚C).

used by researchers for several years to obtain fish species information from eDNA. As shown in Fig 2, the present findings also partially confirmed the reliability of MiFish through the consistency between net sampling data and MiFish fish detection data. On the contrary, considering the simplicity of the procedure and its lower cost, the species-specific PCR method was also used for eDNA when only one or several species were analyzed. Our qPCR data were consistent with the net sampling data. Therefore, as the main focus of this study, we compared the characteristics of qPCR and MiFish methods.

## Quality control of data

Before discussing about the comparison between the characteristics of qPCR and MiFish methods, we must discuss about data quality of this study. Our seawater sampling protocol was relatively old and it was designed based on previous eDNA studies on the marine microorganisms, in which negative controls were not common [39, 40]. Thus, the KS-18-5 samples did not contain the negative controls. However, the later studies have shown the importance of negative controls to monitor possible contamination during the sampling [41, 42]. Therefore, in the cruises after KS-18-5, negative controls by filtering the Milli-Q water during water sampling for OceanDNA were performed. From the negative controls of the seven cruises after

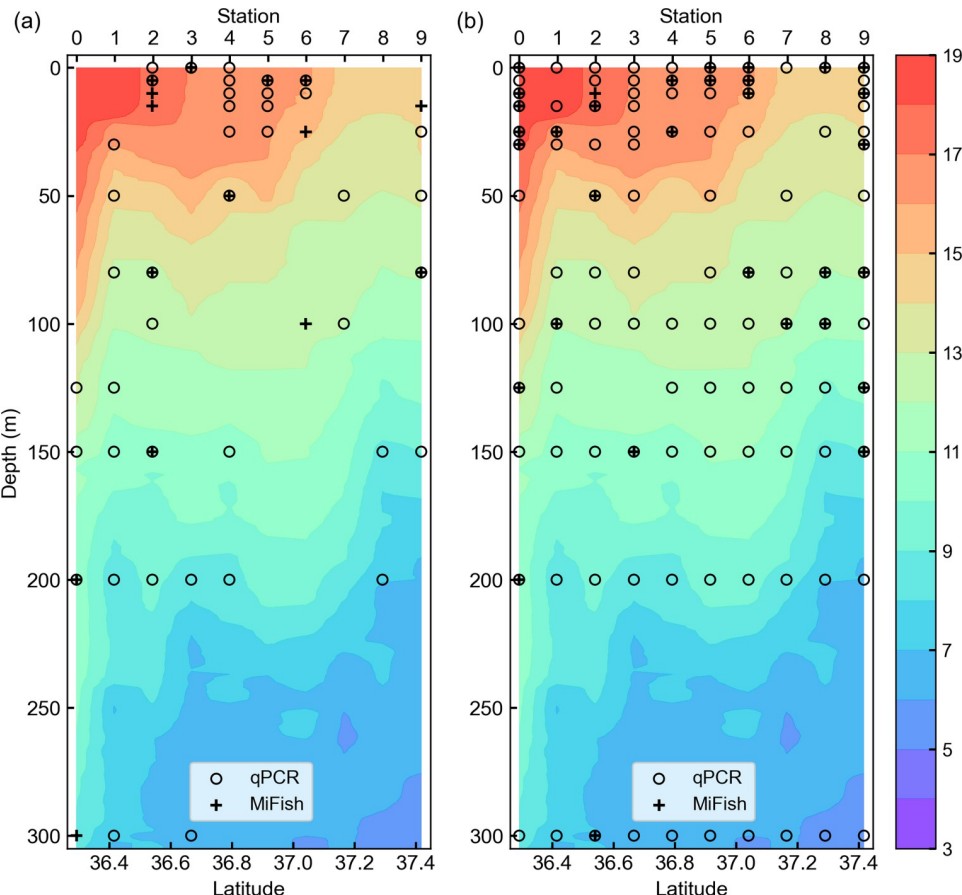

**Fig 6. Comparison of spatial distribution pattern detected by qPCR and MiFish methods in the C-line vertical section.** (a) Spatial distribution of chub/blue mackerel (*Scomber japonicus* and *Sc. australasicus*). (b) Spatial distribution of sardine (*Sardinops melanostictus*). Plus marks represent the presence detection samples found by the MiFish method and open circles represent the presence detection samples found by the qPCR method. Background colors represent water temperature (˚C).

KS-18-5, no target fish DNA have been detected by the qPCR method and no reads have been detected by the MiFish methods (in the form of "library preparation failed as no PCR products obtained"). Although this cannot justify the lack of negative control in KS-18-5, the use of the same seawater sampling method suggested that there could be no contamination in the KS-18-5 sampling.

## Similarity between the detection performance of the qPCR and MiFish methods

In this study, we compared the presence/absence detection results, which represent the existence/absence of the target fish. The results demonstrated the similarity between the detection performance of the qPCR and MiFish methods. First, similar relative abundance rankings among the different target fish were shown using the two eDNA analysis methods. Second, correlation analyses indicated that presence/absence estimations obtained from the qPCR data and the MiFish data always exhibited positive correspondence with each other (Table 1). Based on the combined data of the B- and C- lines, pairwise Phi-coefficients showed positive correlations in four of the five target fish as well as in multiple target pooling data. These results

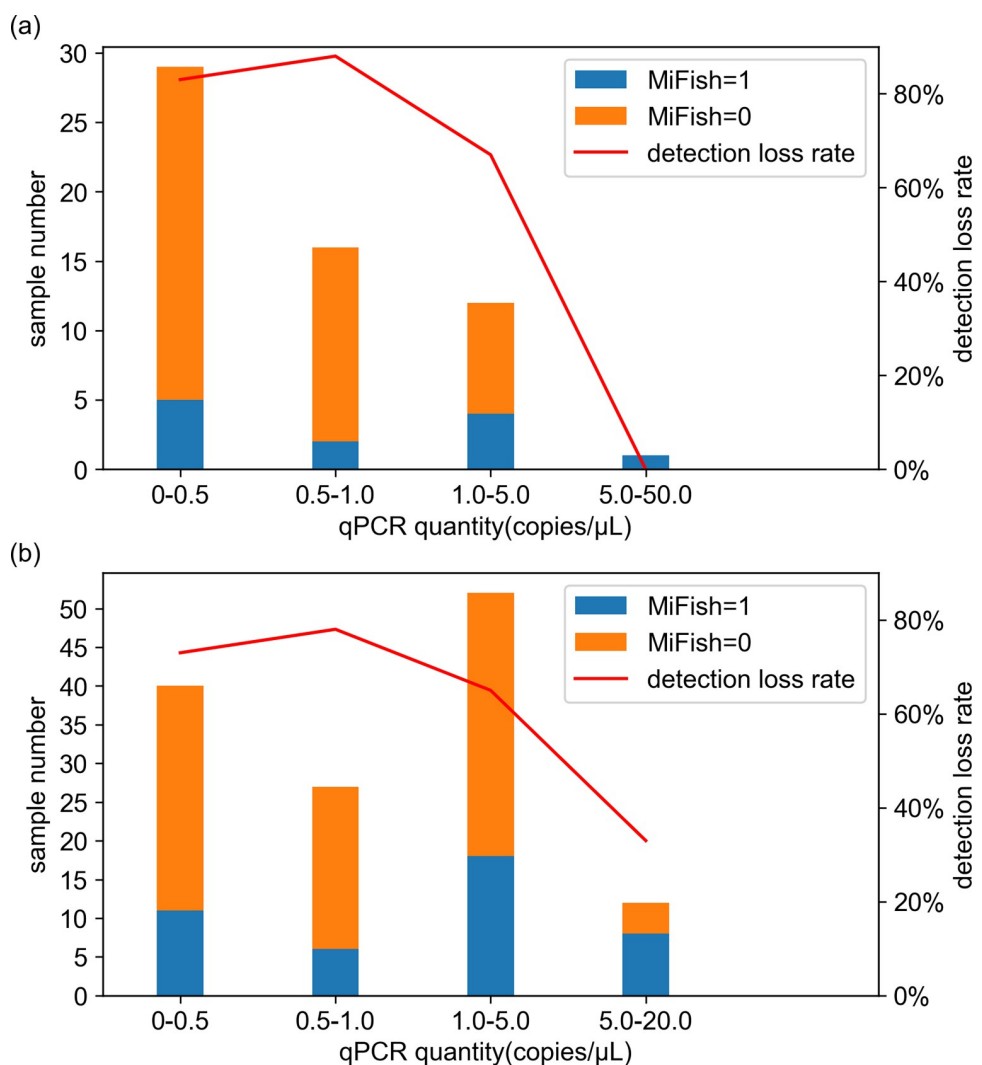

**Fig 7. Comparison of MiFish presence/absence detection samples based on OceanDNA quantity measured by qPCR.** (a) Result of chub/blue mackerel (*Scomber japonicus* and *Sc. australasicus*) and (b) result of sardine (*Sardinops melanostictus*). The figures were prepared from pooling data of B-line and C-line. Samples in which qPCR did not detect OceanDNA are not included in this figure. Numbers in x axis represent the OceanDNA quantity (copies/μL) measured by qPCR.

**Table 2. Difference of non-target OTU number (average per sample) and chlorophyll-a concentration (average per sample) between MiFish-0/qPCR-1 and MiFish-1/qPCR-1 samples.**

| | Non-target OTU number | | | Chlorophyll-a concentration (mg/m³) | | |
|---|---|---|---|---|---|---|
| | chub/blue mackerel | anchovy | sardine | chub/blue mackerel | anchovy | sardine |
| MiFish-0/qPCR-1 | 18.29 | 16.92 | 29.16 | 0.65 | 0.57 | 0.59 |
| MiFish-1/qPCR-1 | 13.58 | 8.00 | 11.53 | 0.59 | 1.35 | 0.90 |
| *P* | <0.001 | <0.001 | <0.001 | <0.001 | <0.001 | <0.001 |

The data were based on samples after excluding library failure samples. MiFish-0/qPCR-1 samples represent samples in which the qPCR method detected the target fish, whereas the MiFish method did not. MiFish-1/qPCR-1 samples represent samples in which both the qPCR and MiFish methods detected the target fish.

suggested that both qPCR and MiFish methods can detect small pelagic fish. Third, a similar spatial distribution pattern was evident between the MiFish and qPCR detection results, especially for B-line sardine and C-line chub/blue mackerel (Figs 5 and 6). The findings showed comparable similarity and positive correlation between the detection performance of the qPCR and MiFish methods.

## Higher detection rate in the qPCR method than in the MiFish method

Despite the similarity and positive correlation shown above, the difference between the detection performance of the qPCR and MiFish methods was also clearly shown in this study. According to the detection results (S7 Table), among the five target fish, the qPCR method showed a much higher detection rate than that of the MiFish method for the three targets, including the most and second most common fish. Combining B-line and C-line data together, the detection rate was 2.48 times for chub/blue mackerel, 2.67 times for sardine, and 3.50 times for anchovy in the qPCR method compared with in the MiFish method. In addition, only one detection of jack mackerel was detected by qPCR.

Previous studies have also shown that species-specific qPCR has a higher sensitivity than eDNA metabarcoding. For example, in a study in Blakney Creek, New South Wales, metabarcoding failed to detect redfin perch (*Perca fluviatilis*) DNA in six of eight samples where qPCR showed a positive result [43]. In two studies in the same survey area, TaqMan qPCR detected catfish (*Silurus glanis*) at two sampling sites and European eel (*Anguilla anguilla*) at one sampling site, whereas eDNA metabarcoding failed to detect these two fish [44, 45]. In addition, previous eDNA survey studies on other aquatic animals also showed a higher detection ability of qPCR than of the metabarcoding method, such as in great crested newt (*Triturus cristatus*) [15] and Mediterranean fanworm *Sabella spallanzanii* [30]. Our study and the aforementioned studies indicate there is a difference in detection sensitivity between the qPCR method and metabarcoding method in the survey of aquatic creatures using eDNA.

## Relationship between OceanDNA quantity and difference in the detection rate between MiFish and qPCR

Including MiFish, eDNA metabarcoding methods have been widely used in wildlife surveys, especially in species diversity surveys [6, 46–48]. The ability to detect a large number of species in a single performance is a great advantage of metabarcoding compared with species-specific PCR. However, as shown above, the lower sensitivity of the eDNA metabarcoding evident in many studies is a problem that researchers need to consider.

To help overcome this problem, we tested some possible reasons for the low detection rate of the MiFish method compared to the qPCR method. The first hypothesis is that the detection limit of the MiFish method is higher than that of the qPCR method. In this case, when the OceanDNA quantity of the target fish is very low, it would still be detectable by qPCR, but very difficult to detect using the MiFish method. Given the wide distribution of these fish in the survey area, we focused on the sardine and chub/blue mackerel test results. Samples with the highest OceanDNA quantity had the lowest MiFish detection loss rate, as expected (Fig 7). However, the MiFish detection loss rate did not change appreciably between samples with 1.0–5.0 copies/μL OceanDNA quantity and lower OceanDNA quantity (Fig 7). Therefore, the presently-observed lower detection rate of the MiFish method compared with that of the qPCR method can only be partially attributed to the lower detection limit of the MiFish method with low eDNA quantity in the open ocean.

## Other possible reasons for difference in detection rate between MiFish and qPCR

Several previous studies have found that amplification bias is a reason for missing some species in eDNA metabarcoding [43, 49–51]. Normally, amplification bias is thought to be the preferential amplification on some DNA, especially high relative abundance DNA [52–55]. Including the MiFish used in this study, the primers used in eDNA metabarcoding focus on universal sequences to cover a wide range of species. Therefore, preferential amplification revealed the existence of species that released less eDNA and were masked by species that left more DNA in the sample [23, 24]. In the present study, in MiFish-0/qPCR-1 samples, the number of non-target OTUs tended to be higher than that in the MiFish-1/qPCR-1 samples (Table 2). However, the chlorophyll-a concentrations did not show such a tendency (Table 2). The chlorophyll-a concentration represents the amount of algae; algae DNA is not the target of MiFish primers, and it should not be amplified (while chlorophyll-a itself inhibits PCR reactions [56]). Therefore, our results also seem to support amplification bias on non-target-species DNA to be a reason for the detection rate difference between the MiFish method and the qPCR method, while inhibition from chlorophyll-a seemed to not be the reason. However, amplification bias cannot completely explain this difference.

## Library preparation failures in MiFish

As mentioned before, library preparation using the MiFish method failed in 63 OceanDNA samples (S3 Table). These samples were not included in the above analyses. If these samples were included, the comparison more obviously revealed the difference in the detection rates between the two methods (Fig 8, S8 Table). The difference in the detection rates between the qPCR and MiFish methods for chub/blue mackerel, anchovy, and sardine increased (Fig 8A), and the detection rate differences also increased from 2.46 to 2.84 for the pooled data (Fig 8B). On the contrary, the Phi-coefficient value between qPCR and MiFish presence/absence data decreased for the pooling data (Fig 8C).

One critical step in the MiFish library preparation is the first-round PCR that the universal PCR primers amplify the target gene fragments across the target taxa [21]. Therefore, the library preparation failure is likely to happen in the samples with virtually no DNA of target taxa. Indeed, in the negative controls we collected during the later cruises, the library preparation also failed, as mentioned earlier. The lack of target taxa DNA is also possible in seawater samples for the high proportion of microbial DNA and the sparse distribution of fish eDNA [10]. Although MiFish primers were reported to be very effective [57, 58], there are still possible problems like PCR dropout which can lead to amplification failures of existing fish DNA in samples [21]. As shown in our study, practical library preparation of OceanDNA could be challenging in the MiFish method.

## Perspectives for future development of OceanDNA analysis methods in marine fish distribution surveys

The present comparison between OceanDNA results and real direct net survey results (Figs 1 and 2; although the net sampling data was limited to one station) confirmed the ability of the qPCR and MiFish methods, when used on OceanDNA, to be useful tools for small pelagic fish distribution surveys. We believe it is valuable to further develop these two OceanDNA analysis methods. The MiFish method has been proven to be a sensitive and effective tool in eDNA studies for fish. However, the present results and those of previous studies also showed that metabarcoding methods, such as MiFish, sometimes suffer a lower detection rate than the

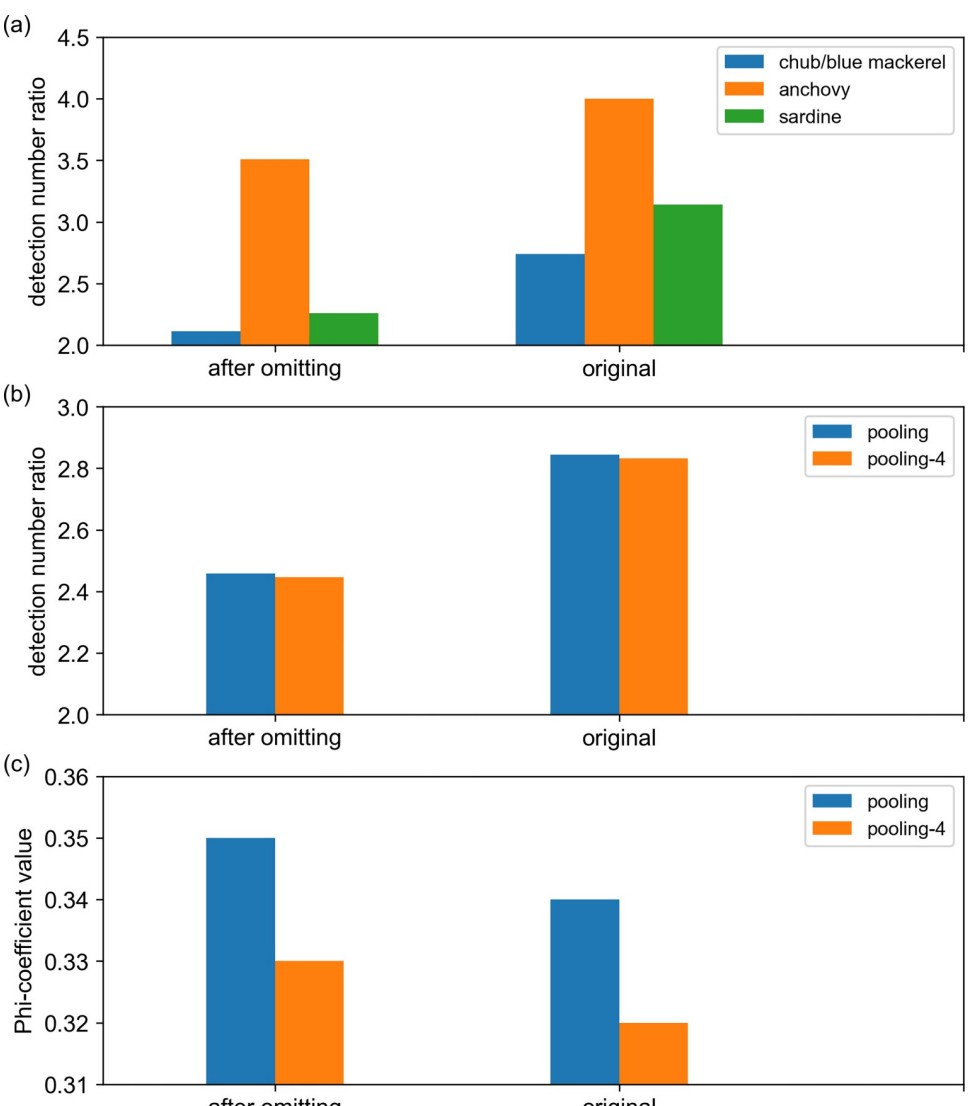

**Fig 8. Differences between the detection results of the qPCR and MiFish methods before and after exclusion of samples in which the library preparation failed.** (a) The detection number ratios (qPCR detection number / MiFish detection number) of chub/blue mackerel (*Scomber japonicus* and *Sc. australasicus*), anchovy (*Engraulis japonicus*) and sardine (*Sardinops melanostictus*). (b) The detection number ratios of pooling and pooling-4 (pooling-4 = sum of chub/blue mackerel, anchovy, sardine and saury, pooling = chub/blue mackerel, sardine, anchovy, saury, and jack mackerel). (c) Phi-coefficient values of pooling and pooling-4. 'original' means the data came from all 247 OceanDNA samples, 'after omitting' means those samples, in which the library preparation failed, were excluded.

qPCR method. In addition, primer efficiency can influence the detection by the eDNA metabarcoding method. For example, in one study, three metabarcoding primers gave different primer efficiencies in silico tests, and 16S rRNA or CytB primers sometimes displayed an efficiency close to 0 [59].Another study also showed that in the same water samples, the metabarcoding method targeting the COI gene primer had a much higher detection rate compared to the 18S rRNA [30]. On the contrary, the results from eDNA metabarcoding cannot be replaced by those from qPCR. The present findings highlight that there are always some unique detection results found only by the MiFish method for each of the five target fish (S7 Table), especially for the detection of saury. Moreover, when conducting research aimed at large numbers

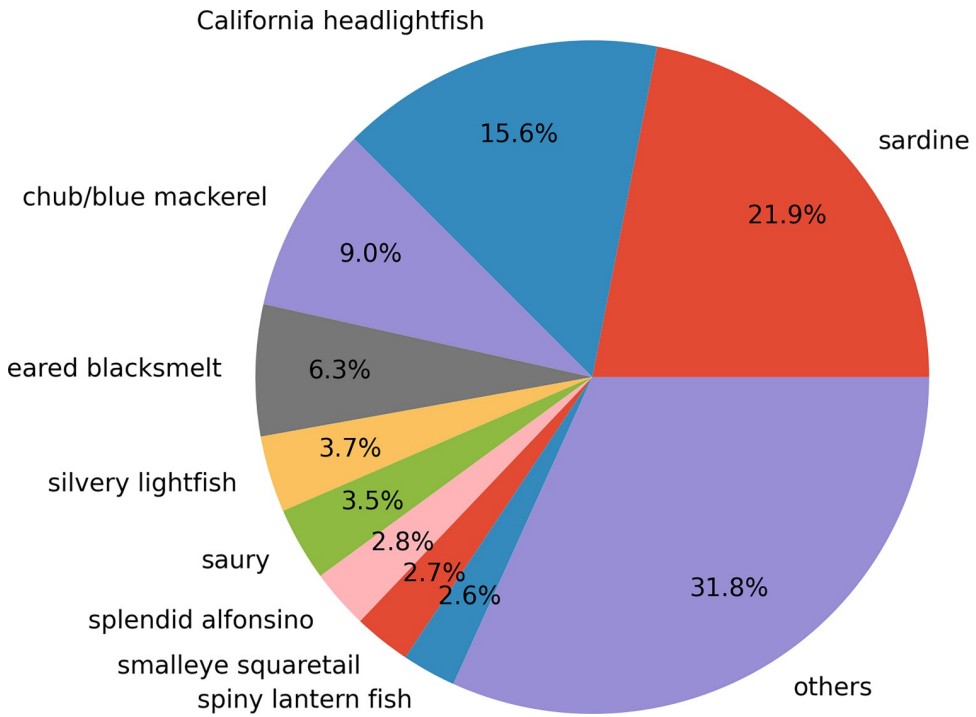

**Fig 9. Pie chart based on read percentages of fish species detected by the MiFish method.** Here, the pie chart was made based on MiFish data from C-line samples that at least one of sardine, chub/blue mackerel and anchovy was detected by the qPCR method, where sardine = *Sardinops melanostictus*, California headlightfish = *Diaphus theta*, chub/blue mackerel = *Scomber japonicus* + *Sc. australasicus*, eared blacksmelt = *Lipolagus ochotensis*, silvery lightfish = *Maurolicus muelleri*, saury = *Cololabis saira*, splendid alfonsino = *Beryx splendens*, smalleye squaretail = *Tetragonurus cuvieri*, spiny lantern fish = *Myctophum spinosum*.

of species, such as biodiversity studies, the use of the eDNA metabarcoding method is essential.

Given the information above, we propose combining the qPCR and MiFish methods in OceanDNA analysis to monitor quantitative distribution of small pelagic fish species with information of fish community structures. The qPCR method provides quantitative estimates and special distribution of target fish species, as shown in Figs 5 and 6. On the other hand, using MiFish data of the water samples in which the qPCR method detected the target species (we called them selected samples), fish community coexisting with the target fish can be inferred. For example, we estimated the fish community coexisting with sardine and chub/blue mackerel (Fig 9). This calculation assumed that DNA quantity is proportional to the MiFish read number over the sum of the samples. At this stage, the MiFish data do not have quantitative information. Therefore, the ratios in Fig 9 do not represent relative abundance. To overcome this issue, quantitative sequencing (qSeq) technique, which adds a random sequence tag to the target sequence before PCR in the library preparation to estimate the DNA quantity from read number found in the metabarcoding [60, 61], is required. However, we could not find any previous study on fish with qSeq in the open ocean. Besides, even with qSeq, there is still a risk that the 'read number-DNA quantity' relationship varies among different species, which can be examined by qPCR data of several target fish. In addition, the added random sequence tag for qSeq consumed some reads in the sequencing, which may reduce the read number we can obtain from the target fish DNA. In our preliminary experiments (data has not been submitted), adding the sequence tag increased the number of non-detected

species in MiFish, which suggested the possible difficulty of qSeq in OceanDNA. Thus, we suggest the combination of the qPCR and MiFish methods as a possible way to detect a large number of species that coexist with the target fish. The combined methods will permit a comprehensive understanding of the quantitative distribution of small pelagic fish within fish communities in the open waters.

## Conclusions

The eDNA analysis methods of species-specific PCR and eDNA metabarcoding are widely used in eDNA survey research [10, 62, 63]. eDNA metabarcoding methods, such as MiFish, can detect a large number of species in one working series, and so have been actively used in research that aims to investigate multiple species [10, 63, 64]. For studies that focus on a single target species, species-specific PCR can also be a good choice because of its simplicity, high sensitivity, and low cost [14]. For OceanDNA research, which is still being developed, the selection of eDNA analysis methods is an important decision for researchers. To provide some clarity for this decision, we conducted an eDNA survey in the open ocean area of the Kuroshio Extension using both the species-specific qPCR and MiFish methods. Five target small pelagic fish (chub/blue mackerel, sardine, anchovy, jack mackerel, and saury) were selected to compare the detection performance of the two eDNA analysis methods. The similar existence/absence detection performance and spatial distribution pattern estimation were evidence of the consistency between the detection results of qPCR and MiFish. At the same time, the obvious detection rate difference between the qPCR method and the MiFish method showed that one method cannot simply replace the other. Our study results indicate that the influence of non-target species DNA on the amplification in MiFish and the lower sensitivity of the MiFish method on lower DNA quantity can partially explain the detection rate difference between the qPCR and MiFish methods. However, more studies are needed to determine the difference between the survey results of the two methods. Thus, we suggest the value of these two OceanDNA analysis methods as well as the necessity for further developing.

## Supporting information

**S1 Text. Detailed procedure for the MiFish method.**
(PDF)

**S1 Fig. The library procedure, thermal cycle settings, and primer sequences of the MiFish method in this study.**
(PDF)

**S2 Fig. Comparison between the spatial distribution of presence detection samples found by MiFish and the eDNA quantity obtained from the qPCR results.**
(PDF)

**S1 Table. Longitude, latitude, and CTD starting time of each sampling station.**
(XLSX)

**S2 Table. Species-specific qPCR primers and probes in this study.**
(XLSX)

**S3 Table. Library preparation results for each sample.**
(XLSX)

**S4 Table. Number of OTU reads in B-line OceanDNA samples.**
(XLSX)

**S5 Table. Number of OTU reads in C-line OceanDNA samples.**
(XLSX)

**S6 Table. qPCR quantity (copies/μL) for all OceanDNA samples.**
(XLSX)

**S7 Table. Detection results of target fish by the qPCR and MiFish methods.**
(XLSX)

**S8 Table. Comparison tables for Phi-coefficients analysis of detection performance.**
(PDF)

**S9 Table. Temperature and chlorophyll-a concentration data recorded in the sampling area.**
(XLSX)

## Acknowledgments

The OceanDNA survey was conducted by the R/V Shinsei-Maru. We thank the captain and all members of cruise KS-18-5. We also appreciate the assistance of Dr. Megumi Enomoto for sea water sampling. We express special thanks to Mr. Shinsuke Toyoda of Marine Work Japan for his operation of the CTD water sampling device. MiFish method analysis was performed with the help of Bioengineering Lab. Co., Ltd.

## Author Contributions

**Conceptualization:** Zeshu Yu, Shin-ichi Ito.

**Data curation:** Zeshu Yu, Shin-ichi Ito, Marty Kwok-Shing Wong, Susumu Yoshizawa, Jun Inoue, Ryuji Yukami, Kazuo Ishikawa, Chenying Guo, Minoru Ijichi.

**Formal analysis:** Zeshu Yu, Marty Kwok-Shing Wong, Minoru Ijichi, Susumu Hyodo.

**Funding acquisition:** Shin-ichi Ito, Susumu Yoshizawa, Sachihiko Itoh, Susumu Hyodo.

**Investigation:** Zeshu Yu, Shin-ichi Ito, Marty Kwok-Shing Wong, Jun Inoue, Kazuo Ishikawa, Chenying Guo.

**Methodology:** Zeshu Yu, Shin-ichi Ito, Marty Kwok-Shing Wong, Susumu Yoshizawa, Sachihiko Itoh, Minoru Ijichi, Susumu Hyodo.

**Project administration:** Shin-ichi Ito, Susumu Hyodo.

**Resources:** Shin-ichi Ito, Susumu Yoshizawa, Sachihiko Itoh, Ryuji Yukami, Susumu Hyodo.

**Software:** Zeshu Yu, Shin-ichi Ito, Marty Kwok-Shing Wong, Susumu Yoshizawa, Minoru Ijichi.

**Supervision:** Shin-ichi Ito, Susumu Hyodo.

**Validation:** Zeshu Yu, Shin-ichi Ito, Marty Kwok-Shing Wong, Susumu Yoshizawa, Jun Inoue.

**Visualization:** Zeshu Yu, Shin-ichi Ito, Marty Kwok-Shing Wong.

**Writing – original draft:** Zeshu Yu, Shin-ichi Ito.

**Writing – review & editing:** Zeshu Yu, Shin-ichi Ito, Marty Kwok-Shing Wong, Susumu Yoshizawa, Jun Inoue, Sachihiko Itoh, Ryuji Yukami, Kazuo Ishikawa, Chenying Guo, Minoru Ijichi, Susumu Hyodo.

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
