## [Decision Letter · Decision Letter 0]

15 Feb 2022

PONE-D-21-36080Comparison of species-specific qPCR and metabarcoding methods to detect small pelagic fish distribution from open ocean environmental DNAPLOS ONE

Dear Dr. Ito,

Thank you for submitting your manuscript to PLOS ONE. After careful consideration, we feel that it has merit but does not fully meet PLOS ONE’s publication criteria as it currently stands. Therefore, we invite you to submit a revised version of the manuscript that addresses the points raised during the review process.

 I have one review which points several corrections and urge the authors to carefully consider them and revise suitably for further consideration

We look forward to receiving your revised manuscript.

Kind regards,

Arga Chandrashekar Anil, Ph. D., D. Agr.,

Academic Editor

PLOS ONE

Journal Requirements:

“SI, JP21H04735, The Japan Society for the Promotion of Science (JSPS) KAKENHI, https://www.jsps.go.jp/english/, NO

SI & SH, The OceanDNA project, The University of Tokyo Future Society Initiative, https://www.u-tokyo.ac.jp/adm/fsi/ja/projects/sdgs/projects_00103.html, NO”

“The contribution of SI was supported by the Japan Society for the Promotion of Science (JSPS) KAKENHI (Grant Number JP21H04735) and the OceanDNA project under the University of Tokyo Future Society Initiative. The OceanDNA survey was conducted by the R/V Shinsei-Maru. We thank the captain and all members of cruise KS-18-5. We also appreciate the assistance of Dr. Megumi Enomoto for sea water sampling.”

“SI, JP21H04735, The Japan Society for the Promotion of Science (JSPS) KAKENHI, https://www.jsps.go.jp/english/, NO

SI & SH, The OceanDNA project, The University of Tokyo Future Society Initiative, https://www.u-tokyo.ac.jp/adm/fsi/ja/projects/sdgs/projects_00103.html, NO”

4. We note that figure 1 in your submission contain [map/satellite] images which may be copyrighted. All PLOS content is published under the Creative Commons Attribution License (CC BY 4.0), which means that the manuscript, images, and Supporting Information files will be freely available online, and any third party is permitted to access, download, copy, distribute, and use these materials in any way, even commercially, with proper attribution. For these reasons, we cannot publish previously copyrighted maps or satellite images created using proprietary data, such as Google software (Google Maps, Street View, and Earth). For more information, see our copyright guidelines: http://journals.plos.org/plosone/s/licenses-and-copyright.

   1. You may seek permission from the original copyright holder of figure 1 to publish the content specifically under the CC BY 4.0 license.  

Maps at the CIA (public domain): https://www.cia.gov/library/publications/the-world- factbook/index.html and https://www.cia.gov/library/publications/cia-maps-publications/index.html

Reviewers' comments:

Reviewer's Responses to Questions

**Comments to the Author**

1. Is the manuscript technically sound, and do the data support the conclusions?

Reviewer #1: Partly

Reviewer #2: Yes

2. Has the statistical analysis been performed appropriately and rigorously? 

Reviewer #1: Yes

Reviewer #2: Yes

3. Have the authors made all data underlying the findings in their manuscript fully available?

Reviewer #1: Yes

Reviewer #2: Yes

4. Is the manuscript presented in an intelligible fashion and written in standard English?

Reviewer #1: No

Reviewer #2: Yes

5. Review Comments to the Author

Reviewer #1: General comments to authors

Authors collected large number of eDNA samples from open ocean environment and performed eDNA analysis using both species-specific qPCR and MiFish metabarcoding. The results indicated both methods have good agreement, while the qPCR provide better detection rates for the target species than the metabarcoding. I consider this study provide important information and knowledge for ocean ecology. However, the one analysis to estimate “corrected” read number is bit confusing (L492-516). In addition, some explanation is insufficient and there are several typos in this MS. I added detailed comments below.

Detailed comments to authors

L26: correct “invasively” to “noninvasively”.

L52-54: Correct to “~the most reliable method. However, the ability of fish influences the catch efficiency and this method requires enormous amount of time in the open ocean, these factors making difficult to obtain high resolution data.”

L61-62: Is this due to open ocean? Reading this paper [11], it seems to be rather due to difficulty in detecting eDNA from Cetaceans. Other studies (e.g., Garcia-Vazquez et al. 2021) on open oceans are more suitable to refer to here.

eDNA metabarcoding of small plankton samples to detect fish larvae and their preys from Atlantic and Pacific waters Eva Garcia-Vazquez, Oriane Georges, Sara Fernandez & Alba Ardura. Scientific Reports volume 11, Article number: 7224 (2021)

L72: “require only a small volume of sample (<10μL). I think this is a minimum volume rather than standard for species specific PCR. You should describe a standard volume of qPCR samples and it is smaller than metabarcoding.

L86-88: Why does this sentence appear in the paragraph of metabarcoding? This does not fit here.

L92: Typo. Correct ”species□specific PCR” to “species-specific PCR.”

L94: Typo. Correct to “However, research comparison….”

L138: Typo. Correct to “An approximately…”

L138: How did you filter 10L of water using a Sterivex? Using aspirator? You should describe the details of procedure.

L155: You should briefly explain the eDNA extraction procedure.

L167: Correct to “the average number of raw and clean reads was 600,718 and 138,476, respectively.”

L172: Don’t refer to an unpublished material.

L191: Add the interpretation of value of Pearon’s Phi-coefficient (e.g. 0.5 or higher means strong positive relationship, 0.3-0.5 means …, 0.1-0.3 mean… ).

L196:Typo. Delete “We”.

L218-220: Add the information regarding body size of the target species base on net sampling.

Fig.3 is not necessary because this information is described in Table1.

L274-: You mentioned that Mifish metabarcoding failed for 63 samples. Even so, did you use the results of all the samples for this analysis?

L274-279: As above mentioned, I don’t know how large the value of Phi-coefficient means a strong relationship between two variables. Please explain this in the statistical analysis section.

L299-322 and Fig.5: It is difficult to distinguish the results from MiFish and qPCR in Fig 5 because both marks are small and overlayed. Enlarge the mark of each mark and add a graph legend for them.

L386-400: Although this analysis is important and valuable for discussion, this paragraph describes the analysis procedure and its results, which should be moved to “Materials and method” and “Results”.

L419: Typo: “hada”?

L416-427 and 444-452: These sentences should be also moved to “Materials and method” and “Results”.

L484: Typo. Correct to ”In the present study”.

L491: Is it really difficult to apply qSeq to Ocean DNA study? Have you ever confirmed? Are there any references?

L496: Fig 9 should be corrected to Fig 8?

L519-520: Grammar error. Correct this sentence.

L487-489: Quantitative MiSeq sequencing (qMiSeq)has been also utilized in marine environments to monitor eDNA concentrations of multiple species simultaneously (Ushio et al. 2018 Metabarcoding and Metagenomics; Ushio 2019 Methods in Ecology and Evolution; Sato et al. 2021 Scientific reports).

L492-516: I do not understand this calculation from your sentences. Why the relationship in Fig. 8 can be true for other samples? MiFish read percentages in this figure may also have amplification bias. In that case, you cannot use this relationship to estimate the correct fish composition. At least, you should use the equation to explain this calculation for easier understanding to readers. In addition, these sentences also include “Materials and method” and “Results” as above mentioned. Distinguish them from “Discussion”.

Reviewer #2: "Environmental DNA (eDNA) is increasingly used to invasively monitor aquatic animals in

27 freshwater and coastal areas."

The word "invasively" in the first line of the abstract is not correct. eDNA is a non-invasive technique compare to the traditional invasive technique. So application of this world here may cause misunderstanding. I recommend to replace this (invasively) with more relative and meaningful world.

6. PLOS authors have the option to publish the peer review history of their article (what does this mean?). If published, this will include your full peer review and any attached files.

Reviewer #1: No

Reviewer #2: **Yes: **Dr. Rose S.F Afzali

---

## [Author Response · Author response to Decision Letter 0]

29 Apr 2022

Response to academic editor and reviewers

We would like to thank the academic editor and reviewers for their encouragement and their devoted efforts to review the manuscript. We have done our best to address the concerns below. We much appreciate the reviewers for your thorough and constructive reviews, which we think have greatly improved the manuscript. Our responses are in blue and red (red color marked what we modified in the manuscript), while the original comments are in black and italicized.

Detailed response to academic editor 

1. Please ensure that your manuscript meets PLOS ONE's style

requirements, including those for file naming. The PLOS ONE style

templates can be found at

<https://journals.plos.org/plosone/s/file?id=wjVg/PLOSOne_formatting_sample_main_body.pdf>

 and

<https://journals.plos.org/plosone/s/file?id=ba62/PLOSOne_formatting_sample_title_authors_affiliations.pdf>

> Thank you very much for your reminding. This time we carefully checked to make the manuscript meets PLOS ONE's style requirements.

“SI, JP21H04735, The Japan Society for the Promotion of Science (JSPS)

KAKENHI, https://www.jsps.go.jp/english/, NO

SI & SH, The OceanDNA project, The University of Tokyo Future Society

Initiative, https://www.u-tokyo.ac.jp/adm/fsi/ja/projects/sdgs/projects_00103.html, NO”

> Thank you for your reminding. Yes, as you said, the funders had no role in the study, this time we included the statement in the Financial Disclosure section as: 

“SI, JP21H04735, The Japan Society for the Promotion of Science (JSPS) KAKENHI, https://www.jsps.go.jp/english/. The funders had no role in study design, data collection and analysis, decision to publish, or preparation of the manuscript.

SI & SH, The OceanDNA project, The University of Tokyo Future Society Initiative,

https://www.u-tokyo.ac.jp/adm/fsi/ja/projects/sdgs/projects_00103.html. The funders had no role in study design, data collection and analysis, decision to publish, or preparation of the manuscript.”

3. Thank you for stating the following in the Acknowledgments Section of

your manuscript:

“The contribution of SI was supported by the Japan Society for the Promotion of Science (JSPS) KAKENHI (Grant Number JP21H04735) and the OceanDNA project under the University of Tokyo Future Society Initiative. The OceanDNA survey was conducted by the R/V Shinsei-Maru.

We thank the captain and all members of cruise KS-18-5. We also appreciate the assistance of Dr. Megumi Enomoto for sea water sampling.”

We note that you have provided additional information within the Acknowledgements Section that is not currently declared in your Funding Statement. Please note that funding information should not appear in the Acknowledgments section or other areas of your manuscript. We will only

publish funding information present in the Funding Statement section of

the online submission form.

“SI, JP21H04735, The Japan Society for the Promotion of Science (JSPS)

KAKENHI, https://www.jsps.go.jp/english/, NO

SI & SH, The OceanDNA project, The University of Tokyo Future Society

Initiative, https://www.u-tokyo.ac.jp/adm/fsi/ja/projects/sdgs/projects_00103.html, NO”

Please include your amended statements within your cover letter; we will

change the online submission form on your behalf.

> We are sorry for this mistake, this time we deleted the funding-related text in the manuscript and stated the Funding Statement again in the Financial Disclosure section.

4. We note that figure 1 in your submission contain [map/satellite] images which may be copyrighted. All PLOS content is published under the Creative Commons Attribution License (CC BY 4.0), which means that the manuscript, images, and Supporting Information files will be freely available online, and any third party is permitted to access, download, copy, distribute, and use these materials in any way, even commercially, with proper attribution. For these reasons, we cannot publish previously

copyrighted maps or satellite images created using proprietary data, such as Google software (Google Maps, Street View, and Earth). For more information, see our copyright guidelines:

http://journals.plos.org/plosone/s/licenses-and-copyright

<http://journals.plos.org/plosone/s/licenses-and-copyright>.

 We require you to either (1) present written permission from the

copyright holder to publish these figures specifically under the CC BY

4.0 license, or (2) remove the figures from your submission:

 1. You may seek permission from the original copyright holder of figure 1 to publish the content specifically under the CC BY 4.0 license.

We recommend that you contact the original copyright holder with the Content Permission Form

(http://journals.plos.org/plosone/s/file?id=7c09/content-permission-form.pdf

<http://journals.plos.org/plosone/s/file?id=7c09/content-permission-form.pdf>)

and the following text:

“I request permission for the open-access journal PLOS ONE to publish XXX under the Creative Commons Attribution License (CCAL) CC BY 4.0

(http://creativecommons.org/licenses/by/4.0/

<http://creativecommons.org/licenses/by/4.0/>). Please be aware that this license allows unrestricted use and distribution, even commercially, by third parties. Please reply and provide explicit

written permission to publish XXX under a CC BY license and complete the attached form.”

 In the figure caption of the copyrighted figure, please include the following text: “Reprinted from [ref] under a CC BY license, with permission from [name of publisher], original copyright [original

copyright year].”

The following resources for replacing copyrighted map figures may be

helpful:

 USGS National Map Viewer (public domain):

http://viewer.nationalmap.gov/viewer/

<http://viewer.nationalmap.gov/viewer/>

The Gateway to Astronaut Photography of Earth (public domain):

http://eol.jsc.nasa.gov/sseop/clickmap/

<http://eol.jsc.nasa.gov/sseop/clickmap/>

Maps at the CIA (public domain):

https://www.cia.gov/library/publications/the-world- factbook/index.html

<https://www.cia.gov/library/publications/the-world-%20factbook/index.html>

and

https://www.cia.gov/library/publications/cia-maps-publications/index.html <https://www.cia.gov/library/publications/cia-maps-publications/index.html>

NASA Earth Observatory (public domain):

http://earthobservatory.nasa.gov/ <http://earthobservatory.nasa.gov/>

<http://landsat.visibleearth.nasa.gov/>

USGS EROS (Earth Resources Observatory and Science (EROS) Center)

(public domain): http://eros.usgs.gov/# <http://eros.usgs.gov/>

<http://www.naturalearthdata.com/>

> Thank you very much for your statement about the copyright concerns. When drawing the map in Fig 1, we used the data of Natural Earth. But as stated on the website of Natural Earth (https://www.naturalearthdata.com/about/terms-of-use/) as follows, they don’t need users to ask them for permission. 

“All versions of Natural Earth raster + vector map data found on this website are in the public domain. You may use the maps in any manner, including modifying the content and design, electronic dissemination, and offset printing. The primary authors, Tom Patterson and Nathaniel Vaughn Kelso, and all other contributors renounce all financial claim to the maps and invites you to use them for personal, educational, and commercial purposes.

No permission is needed to use Natural Earth. Crediting the authors is unnecessary.

However, if you wish to cite the map data, simply use one of the following.

Short text:

Made with Natural Earth.

Long text:

Made with Natural Earth. Free vector and raster map data @ naturalearthdata.com.”

Thus, we added the statement as “Maps were made in Spyder (Python 3.7) using Natural Earth data. Free vector and raster map data @ naturalearthdata.com.” in the caption of Fig 1.

When drawing Fig 1, we also used the current velocity data (Ocean Surface Current Analysis Real-time) and sea surface temperature data (GHRSST Level 4 OSTIA Global Foundation Sea Surface Temperature Analysis) downloaded from the PO.DAAC. But as stated on the website of PO.DAAC (https://podaac.jpl.nasa.gov/CitingPODAAC) as follows, they also don’t need users to ask them for permission.

“Data hosted by the PO.DAAC is openly shared, without restriction, in accordance with NASA's Earth Science program Data and Information Policy.

Bibliographic citations should be included in the References section of publications and other media to acknowledge those who have created the data, services, and tools provided by the PO.DAAC. Proper citations include the authors, title, publisher, and Digital Object Identifier (DOI) and will allow the products to be discovered and re-used by others. The proper citation for each PO.DAAC product is provided on its landing page and user guide.

Example Data Citation

…”

Thus, we modified the statement as “(a) Colors represent current speeds and arrows represent current velocity vectors on the sea surface (5-day mean from May 10 to 14, 2018) from Ocean Surface Current Analysis Real-time data [34] (hosted and openly shared by the PO.DAAC, without restriction, in accordance with NASA's Earth Science program Data and Information Policy). (b) The contours and colors represent sea surface temperature (℃) distribution on May 10, 2018, from GHRSST Level 4 OSTIA Global Foundation Sea Surface Temperature Analysis data [35] (hosted and openly shared by the PO.DAAC, without restriction, in accordance with NASA's Earth Science program Data and Information Policy)” in the caption of Fig 1. And we also added the Citation in the “References” as the follows, like the PO.DAAC website asked for. 

“34. ESR. 2009. OSCAR third deg. Ver. 1. PO.DAAC, CA, USA. Dataset accessed [2021-07-08] at https://doi.org/10.5067/OSCAR-03D01.

35. UK Met Office. 2005. OSTIA L4 SST Analysis. Ver. 1.0. PO.DAAC, CA, USA. Dataset accessed [2021-07-06] at https://doi.org/10.5067/GHOST-4FK01.”

And finally, we wish to keep the original Fig 1 in the manuscript with the added citation information showed above.

Detailed response to reviewers

Reviewer #1: 

General comments to authors

Authors collected large number of eDNA samples from open ocean environment and performed eDNA analysis using both species-specific qPCR and MiFish metabarcoding. The results indicated both methods have good agreement, while the qPCR provide better detection rates for the target species than the metabarcoding. I consider this study provide important information and knowledge for ocean ecology. However, the one analysis to estimate “corrected” read number is bit confusing (L492-516). In addition, some explanation is insufficient and there are several typos in this MS. I added detailed comments below. 

> Thank you very much for your comment and suggestion. We are sorry that we didn’t make it clear of explaining the MiFish read number adjustment and some other points. This time we added explanation about how we did the “quantifying adjustment of MiFish reads result with qPCR data” in the Materials and methods. We also added some other explanations (such as the explanation of water filtering method and the explanation of eDNA purification method) according to your suggestion. We deeply apologize for the typos, this time we made it more carefully in checking the MS to clean up the typographical errors. 

Detailed comments

L26: correct “invasively” to “noninvasively”.

>We deeply apologize for the typo, we have changed the “invasively” to “noninvasively” (L26, line number is based on 'Revised Manuscript with Track Changes' file at "Hidden revisions" status for here and below).

L52-54: Correct to “~the most reliable method. However, the ability of fish influences the catch efficiency and this method requires enormous amount of time in the open ocean, these factors making difficult to obtain high resolution data.”

> Thank you very much for your advice. Wave changed the sentence based on your advice: “~the most reliable method. However, the escaping ability of fish influences the catch efficiency, and this method requires an enormous amount of time in the open ocean, which makes it difficult to obtain high resolution data.” (L50-53).

L61-62: Is this due to open ocean? Reading this paper [11], it seems to be rather due to difficulty in detecting eDNA from Cetaceans. Other studies (e.g., Garcia-Vazquez et al. 2021) on open oceans are more suitable to refer to here.

> Thank you for the advice. As you mentioned it seems that the failure of detecting killer whale here may be due to the difficulty in detecting eDNA from Cetaceans. Therefore, we have changed the Róisín Pinfield et al. 2019 to the Garcia-Vazquez et al. 2021 which showed the difficulty of for detecting planktons in open ocean compared with coastal area for the reference [11]. We also changed the “fish and marine mammals” to “fish and zooplankton” (L59) and changed “in one study, although water samples were collected very near offshore killer whales (Orcinus orca), killer whale eDNA was not detected” to “For example, zooplankton were detected in the coastal region, but not in the open waters, in 1.5 L water samples using COI metabarcoding” (L60-61).

(Garcia-Vazquez et al. 2021, doi: 10.1038/s41598-021-86731-z.)

L72: “require only a small volume of sample (<10μL). I think this is a minimum volume rather than standard for species specific PCR. You should describe a standard volume of qPCR samples and it is smaller than metabarcoding.

> Thank you for your suggestion. Standard volume required for one qPCR reaction is about 2-2.5 μL (Minegishi et al. 2019, Takahara et al. 2013) and standard volume required for one reaction in metabarcoding varies among different studies (2μL in Markus et al. 2018, 10μL in Nathan et al. 2015). As sometimes required volume of metabarcoding is as small as qPCR, we decided to delete the sentence “require only a small volume of sample (<10μL)” when explaining advantages of qPCR (L71).

(Minegishi et al. 2019, doi: 10.1371/journal.pone.0222052.

Takahara et al. 2013, doi: 10.1371/journal.pone.0056584.

Markus et al. 2018, doi: 10.1038/s41598-018-23052-8.

Nathan et al. 2015, doi: 10.1111/1755-0998.12433.)

L86-88: Why does this sentence appear in the paragraph of metabarcoding?

This does not fit here.

> Sorry for causing the misunderstanding here. In this part, we wanted to introduce the disadvantage of both qPCR and metabarcoding here. To make it clear, we moved the introduction of MiFish (“For fish species, a new eDNA metabarcoding … in the eastern equatorial Pacific [19, 21].”) to the previous paragraph that introduced the qPCR and eDNA metabarcoding and their advantages. Then, we changed the remaining part in this paragraph to “However, both qPCR and eDNA metabarcoding face observational biases due to the degradation of eDNA, advection by ocean currents, and the inhibition of DNA amplification by additional substances [22-24]. In addition, amplification bias of high-throughput sequencing (HTS) can influence the sensitivity of eDNA metabarcoding [25-28]. High abundance of eDNA may inhibit the amplification of low-abundance eDNA through the competition of binding with metabarcoding primers [25-27]. Both issues can result in the loss of some species in eDNA metabarcoding analysis.” to introduce the disadvantage of both qPCR and metabarcoding here (L81-86). 

L92: Typo. Correct ”species□specific PCR” to “species-specific PCR.”.

>Sorry for the special character, we have corrected it (L90-91). 

L94: Typo. Correct to “However, research comparison….”

>Sorry for our mistake, we have corrected it (L92).

L138: Typo. Correct to “An approximately…”

>Sorry for our mistake, we have corrected it (L136). And we are very sorry that we made another mistake here. In fact, we filter approximately 7L water for each sample in this study, so we finally corrected here as “Approximately 7 L…” (L136).

L138: How did you filter 10L of water using a Sterivex? Using aspirator?

You should describe the details of procedure.

> Thank you for your suggestion. Now we added the details to show the procedure of water filtering here (L137-145).

“To perform filtering, inlet end of the Sterivex-GP pressure filter unit was attached to the 1/4 inch HB to M Luer lock (XX3002564; Merck Biopharma Co., Ltd., Tokyo, Japan), which was assembled into one end of a peroxide-cured silicon pump tube (L/S25, 96400-25; Yamato Scientific Co., Ltd., Tokyo, Japan). The pump tube was then fixed by tube cartridge (07519-70; Yamato Scientific Co., Ltd.) to the multi-channel pump head (07519-06; Yamato Scientific Co., Ltd.). The pump head was assembled into a digital pump (07528-10; Yamato Scientific Co., Ltd.). Finally, through the peroxide treatment silicon pump tube, the digital pump (rotation speed set to 60 rpm) pushed the water sample through the Sterivex-GP pressure filter unit from the plastic bag.”

L155: You should briefly explain the eDNA extraction procedure.

> Thank you for your suggestion. Now we added some explanation of eDNA extraction and purification here (L163-170).

“DNA extraction was performed using Charge Switch Forensic DNA Purification Kit (Thermo Fisher Scientific). Briefly, after removing RNAlater, 2 mL lysis mix (containing Lysis Buffer and 20 µL Proteinase K) was added to each Sterivex filter unit. The filter units were incubated at 55℃ for 30 min. Supernatant from each filter unit was collected in a new microcentrifuge tube. ChargeSwitch® Magnetic Beads (Thermo Fisher Scientific) were added to bind to the DNA. After washing the magnetic beads with Wash Buffer, DNA combined with Magnetic Beads was eluted with 150 µL of Elution Buffer for each sample and then transferred to a new microcentrifuge tube.”

L167: Correct to “the average number of raw and clean reads was 600,718 and 138,476, respectively.”

>Sorry for our mistake, we have corrected it (L177-178). 

L172: Don’t refer to an unpublished material.

>As this article is already published now, we included it in the references list as [36] this time (L181).

36. Marty Kwok-Shing Wong, Shigenori Nobata, Shin-ichi Ito, Susumu Hyodo. Development of species-specific multiplex real-time PCR assays for tracing the small pelagic fishes of North Pacific with environmental DNA. Environ DNA. 2022 January 14. doi: 10.1002/edn3.275.

L191: Add the interpretation of value of Pearon’s Phi-coefficient (e.g. 0.5 or higher means strong positive relationship, 0.3-0.5 means …, 0.1-0.3 mean… )..

>Thank you for your suggestion. We added some interpretation of Phi-coefficient value. However, as precise meaning of coefficient value depends on the freedom of data, we more depended on the p-value (Fisher's exact test) here rather than using “crude estimates” coefficients meaning list. (L211-214)

“The value of the Pearson’s Phi-coefficient ranges from -1 to +1, where +1 (-1) indicates perfect agreement (disagreement) and 0 indicates no relationship. As the precise meaning of the coefficients (strengths of relationships) depends on the freedom of data, we also calculated the p-value of Fisher's exact test.”

L196:Typo. Delete “We”.

>Sorry for the typo, we have corrected it (L197). 

L218-220: Add the information regarding body size of the target species base on net sampling.

>Thank you for the suggestion. We added the information of the body size of the target species collected by the net sampling here (L265-268). 

“The target fish species caught in the net comprised 158 sardine (40.85-66.80 mm body length, 0.85-3.74 g body weight; 0.246 kg in total), 289 chub/blue mackerel (16.99-64.08 mm fork length. 0.04-2.75 g in body weight; 0.044 kg in total), and 40 anchovies (34.99-68.68 mm body length, 0.35-3.24 g in body weight; 0.044 kg in total).”

Fig.3 is not necessary because this information is described in Table1.

>Thank you for your advice. As you said, the information of Fig 3 is also included in the Table 1. However, when comparing the detection rate from qPCR and from MiFish among five target fish, we think the figure version will be clearer to understand. So, here we kept the Fig 3 (L300) and to prevent the information repetition, we moved the original Table 1 to the Supporting information part as S7 Table. 

L274-: You mentioned that Mifish metabarcoding failed for 63 samples. Even so, did you use the results of all the samples for this analysis?

>Thank you for your question. By your question, we realized that we made a mistake. We did not include the 63 samples that failed in this Phi-coefficient examination, thus we corrected “247 OceanDNA samples” to “184 OceanDNA samples” here (L312).

L274-279: As above mentioned, I don’t know how large the value of Phi-coefficient means a strong relationship between two variables. Please explain this in the statistical analysis section..

>As our answers for your question to original L191, we have added some interpretation of Phi-coefficient value, but we more depended on the p-value (Fisher's exact test) here. From the p-value we can see statistically significant positive correlations were found for chub/blue mackerel, sardine, and saury as well as for pooled data (L316-320).

L299-322 and Fig.5: It is difficult to distinguish the results from MiFish and qPCR in Fig 5 because both marks are small and overlayed. Enlarge the mark of each mark and add a graph legend for them.

>Thank you for suggestion. We divided the original Fig 5 into two figures, Figs 5 and 6, for PLOS limited the max size of one single figure. Now the figures and the marks are larger, and we also added the legend of qPCR and MiFish. We hope the figures now are enough for reading the detection results from MiFish and qPCR (L342-354).

L386-400: Although this analysis is important and valuable for discussion, this paragraph describes the analysis procedure and its results, which should be moved to “Materials and method” and “Results”.

>Thank you for suggestion. We put the detailed procedure in the “Materials and method” (L221-230). 

“To find the possible reasons for the different detection rate of the qPCR and MiFish methods, we tested two hypotheses. One hypothesis posited that the detection limit of the qPCR method is lower than that of the MiFish method. The other hypothesis posited that OceanDNA amplification of target species is inhibited by OceanDNA amplification of non-target species (amplification bias) in the MiFish method. To test whether the two methods have different DNA quantity detection limits, we divided the samples in which qPCR detected the target species into different groups according to the OceanDNA quantity of target species measured by qPCR and then compared the MiFish detection loss rate among all groups. The MiFish detection loss rate was defined as the percentage of samples in which MiFish failed to detect target species.” 

And then, as you suggested, we built a part as “Influences of detection limit and amplification bias on detection rate” inside “Results” part to include the analyzing results (L370-387). Besides, we did some slight modification in the “Discussion” (L465-471).

L419: Typo: “hada”?

>Sorry for our mistake, we have corrected “hada tended” to “tended to be” (L479). 

L416-427 and 444-452: These sentences should be also moved to “Materials and method” and “Results”.

>Thank you for your comment. We moved the explanation of methods in the “Materials and method” (L230-237).

“~To test the influence of the amplification bias in the MiFish method, we divided the samples in which qPCR detected the target species into the group in which target fish was not detected by MiFish (MiFish-0/qPCR-1 samples) and the group in which MiFish also detected the target fish (MiFish-1/qPCR-1 samples). We then compared the number of non-target OTUs and the chlorophyll-a concentration between these two groups using the Mann-Whitney U test to test the statistical significance. Non-target OTUs were OTUs that belonged to the non-target fish. Chlorophyll-a concentration was included for comparison because it represents the algae quantity, and algae DNA could not be amplified using the MiFish method.”

As for the analysis results in original L416-427, as we explained in the responses to the L386-400 question, we built a part inside “Results” to include them (L387-400). We also did some slight modification in the “Discussion” (such as in L480-486). 

As for original L444-452, like we introduced in the “Materials and method” (S1 Text) the library preparation is the first step of the MiFish procedure. So, the MiFish method cannot be applied to samples in which the library preparation method failed. Till now, the reason of library preparation failure is undermined. Therefore, we thought it is better not to include it in “Results” part. However, library preparation failure did influence the detection ability of MiFish. Then, we included this part in “Discussion” to remind other researchers of this unsolved problem in MiFish method. 

L484: Typo. Correct to ”In the present study”.

>Sorry for the typo. We have modified this sentence. (L532-535)

L491: Is it really difficult to apply qSeq to Ocean DNA study? Have you ever confirmed? Are there any references?

>Thank you for the question. When searching for the previous researches on fish metabarcoding, we didn’t find the record of using qSeq in open ocean eDNA samples. As you mentioned latter, several researchers succeeded performed qSeq for marine fish eDNA, but what they have in common is their samples were collected in coastal area, such as in the bay. For example, in Ushio et al. 2018, their marine eDNA samples were collected in coastal marine ecosystem in Maizuru Bay. We could not find any qSeq studies on open ocean fish. Therefore, we considered that it may be difficult to apply qSeq to OceanDNA when studying on fish (which was collected far away from the land, just as the Kuroshio Extension where we collected our samples). However, there is one research that performed qSeq in open ocean successfully for microorganisms (Lin et al. 2019), so we agreed that our original statement here is not proper enough.

 To make the explanation proper and clear, we modified it as follows (L535-540). 

“qSeq has also been used in eDNA studies on marine fish, but these studies were performed in environments close to the coast or in the bay [54]. For samples collected in open ocean far away from the land, such as the samples in this study, which were collected in the Kuroshio Extension, qSeq application may be difficult for fish study. Indeed, we could not find any previous study on fish with qSeq in the open ocean, although there has been a successful attempt with the microbiome community [55].”

(Ushio et al. 2018, doi: 10.3897/mbmg.2.23297.

Lin et al. 2019, doi: 10.1128/AEM.02634-18.)

L496: Fig 9 should be corrected to Fig 8?

>Thank you for the reminding. Sorry that we made a mistake in citing figure here. And because of the manuscript modification, now Fig 9 here is correct (L545). 

L519-520: Grammar error. Correct this sentence.

>Thank you for your comment. Now we made it clear as follows (L567-569). 

“MiFish read and qPCR quantity in this figure used the data of chub/blue mackerel (Scomber japonicus and Sc. australasicus) in the B-line, chub/blue mackerel in the C-line, anchovy (Engraulis japonicus) in the B-line, anchovy in the C-line, sardine (Sardinops melanostictus) in the B-line, and sardine in the C-line.”

L487-489: Quantitative MiSeq sequencing (qMiSeq)has been also utilized in marine environments to monitor eDNA concentrations of multiple species simultaneously (Ushio et al. 2018 Metabarcoding and Metagenomics; Ushio 2019 Methods in Ecology and Evolution; Sato et al. 2021 Scientific reports).

> Thank you very much for your supplementary introduction. As you mentioned, there are successful usage of Quantitative MiSeq sequencing (qMiSeq) in marine eDNA study. However, as mentioned above, rather than in the open ocean area away from the land, their sampling works are all performed in the areas close to the land, like in the coastal marine ecosystem (Ushio et al. 2018 and 2019) or around artificial reefs in Tateyama Bay (Sato et al. 2021). In fact, we also agree with the worth of trying qMiSeq in open ocean, but it may be difficult. So, we showed another possible way, to combine the use of qPCR and MiFish methods to achieve quantitative sequencing. To make it clear, we modified the text (L536-541). 

L492-516: I do not understand this calculation from your sentences. Why the relationship in Fig. 8 can be true for other samples? MiFish read percentages in this figure may also have amplification bias. In that case, you cannot use this relationship to estimate the correct fish composition. At least, you should use the equation to explain this calculation for easier understanding to readers. In addition, these sentences also include “Materials and method” and “Results” as above mentioned. Distinguish them from “Discussion”.

>Thank you for the question and advice. To make the statement clearer, we added the explanation how we did the quantifying adjustment of MiFish reads result with qPCR data as follows in the “Materials and method” (L238-256).

“Considering the ability of MiFish to detect large numbers of species and the quantitative nature of qPCR, we attempted to combine the two methods to simultaneously determine the quantitative distribution of small pelagic fish species within fish community structures. We wanted to convert the OceanDNA read percentages detected by MiFish to OceanDNA quantities as an average view of the observation vertical sections. First, we investigated whether the MiFish read percentage (y) of the target species and qPCR quantity (x) showed a positive correlation. We used the data of sardine, anchovy, and chub/blue mackerel in the B- and C-line, since their detection rates were high (see Results). The MiFish read percentage indicates the selected species read percentage to the total OTU reads of all B-or C-line samples. Linear regressions were performed using statsmodels.regression.linear_model.OLS of Python. If a significant positive linear correlation between the MiFish read percentage (y) and qPCR quantity (x) was found, the adjusted DNA quantity of species n of all selected samples, Σ(qn), was calculated as:

 Σ〖(q〗_n)=Σ[〖 y〗_n*(q_sardine+q_anchovy+q_mackerel)/(y_sardine+y_anchovy+y_mackerel)],

where qsardine, qanchovy, and qmackerel are the qPCR quantities for sardine, anchovy, and chub/ blue mackerel, respectively; and ysardine, yanchovy, and ymackerel are the MiFish read percentages of sardine, anchovy, and chub/ blue mackerel, respectively. The selected sample was one in which at least one of the sardine, anchovy and chub/blue mackerel was detected by both the MiFish and qPCR methods. The fish species composition was calculated using Σqn and compared with the original composition.”

In our samples, we found a significant positive linear correlation between the MiFish read percentage and the qPCR DNA quantity (L548). Using the method above, we calculated the adjusted DNA quantity from MiFish read number of each fish species. We apologize for causing the misunderstanding, but the method here was not for solving amplification bias problem. What we tried to do is estimating the DNA quantity from MiFish read number. 

As you pointed out, the linear MiFish-qPCR correlation we got here was based on data in this study, so may not apply to other eDNA studies. In fact, we did not intend that we discovered a universal quantitative relationship between qPCR and MiFish data. What we wanted to indicate is a possible way of combining usage of MiFish and qPCR. Other researchers can simultaneously quantify and sequence multiple fish species if they discover the MiFish-qPCR quantitative relationship in their own OceanDNA samples. 

In this study, we studied about the validity of each eDNA analysis methods (represented by MiFish and qPCR) to help developing eDNA method for fish distribution survey in open ocean. After a series of study, we confirmed the validity of both methods but also recognized the lower detection rate of MiFish method. Although we also tested some possible reasons for lower detection rate of MiFish, our results were still not enough for overcoming this problem. On the other hand, the MiFish and other metabarcoding methods have the irreplaceable capability in multi-species analysis. Thus, it is difficult to choose one method as the “best” eDNA analysis method. So, in the final part of “Discussion”, we wanted to introduce this new idea, combining the usage of qPCR and MiFish, to show a new possible way for open ocean fish survey study in the future. Thus, we wished to keep this part in the position like in the original manuscript. But, to make our purpose clearer, we also did some modification (L530-532, 562-564).

Reviewer #2: "Environmental DNA (eDNA) is increasingly used to invasively monitor aquatic animals in 27 freshwater and coastal areas." The word "invasively" in the first line of the abstract is not correct.

eDNA is a non-invasive technique compare to the traditional invasive technique. So application of this world here may cause misunderstanding. I recommend to replace this (invasively) with more relative and meaningful world.

>Thank you for the comment. We are sorry for the typo here. We have replaced it with “noninvasively” (L26).

---

## [Decision Letter · Decision Letter 1]

20 May 2022

PONE-D-21-36080R1Comparison of species-specific qPCR and metabarcoding methods to detect small pelagic fish distribution from open ocean environmental DNAPLOS ONE

Dear Dr. Ito,

Thank you for submitting your manuscript to PLOS ONE. After careful consideration, we feel that it has merit but does not fully meet PLOS ONE’s publication criteria as it currently stands. Therefore, we invite you to submit a revised version of the manuscript that addresses the points raised during the review process. One of the reviewer has raised several questions related to methodology and deserve careful attention in the revision .

We look forward to receiving your revised manuscript.

Kind regards,

Arga Chandrashekar Anil, Ph. D., D. Agr.,

Academic Editor

PLOS ONE

Reviewers' comments:

Reviewer's Responses to Questions

**Comments to the Author**

1. If the authors have adequately addressed your comments raised in a previous round of review and you feel that this manuscript is now acceptable for publication, you may indicate that here to bypass the “Comments to the Author” section, enter your conflict of interest statement in the “Confidential to Editor” section, and submit your "Accept" recommendation.

Reviewer #1: (No Response)

Reviewer #2: All comments have been addressed

2. Is the manuscript technically sound, and do the data support the conclusions?

Reviewer #1: Yes

Reviewer #2: Yes

3. Has the statistical analysis been performed appropriately and rigorously? 

Reviewer #1: Yes

Reviewer #2: Yes

4. Have the authors made all data underlying the findings in their manuscript fully available?

Reviewer #1: Yes

Reviewer #2: Yes

5. Is the manuscript presented in an intelligible fashion and written in standard English?

Reviewer #1: Yes

Reviewer #2: Yes

6. Review Comments to the Author

Reviewer #1: General comments to authors

Authors addressed the reviewer’s comments and revised the manuscript, which got largely improved. Meanwhile, I noticed that this paper does not contain information about negative controls of eDNA samples (i.e., filtering of distilled water or Milli-Q). Because sampling negative controls is important to check possible contamination during field and laboratory processes, authors should add sentences about it. Besides, I do not consider that the combining method of metabarcoding and qPCR results of the target species can provide reliable estimates of eDNA quantity of the non-target species (L238-256, L529-564 and 603-605). Therefore, I recommend authors to delete the related sentences and figures about this method. I provided detailed comments about these points, as well as minor comments below.

Detailed comments to authors

L62 The sentence of “The foregoing indicates that~” should be included in the previous paragraph.

L108 Please add information of “scarce eDNA in open ocean” as “~ in the open ocean, where the concentration of fish eDNAs are expected to be scarce, has been insufficient.“

This will increase a necessity for this study

L151 Correct to “~were collected from water depths of 5 to 300 m”.

L131-177 I noticed that this paper does not mention about negative controls of eDNA samples (i.e., filtering of distilled water or Milli-Q). Sampling negative controls is necessary to check possible contamination during field and laboratory processes. Please add sentences about it.

L229-232 You should decide this detection threshold of metabarcoding based on the result of negative controls (Yamamoto et al. 2016; Sato et al. 2021). At least, you should show read number of negative controls.

L238-256 I understand your calculation for the adjusted DNA quantity of species n of all selected samples. However, I do not consider this calculation reliably estimate DNA quantity of the non-target species of qPCR because MiFish read percentage can vary with read number of other species as shown around 20 copies/μl of qPCR quantity in figure 9. MiFish read percentage can increase or decrease without change of its DNA quantity or read number if read numbers of other species change low or high, respectively. I consider such variability of MiFish read percentage can prevent from estimating reliable DNA quantity of non-target species. Therefore, I recommend authors delete the sentences about this method and Figure 9-11.

L262 Please add information about a depth range or depth layer (bottom or middle?) of net sampling in the main body and caption of Fig.2.

L325 and Fig.4 Checking figure 4, explanation in the caption seems to be incorrect. Pooling-4 and pooling should correspond to (f) and (e), respectively.

L488 In the discussion, you should infer possible reasons for preparation failures in MiFish libraries.

L492-494 How could you still calculate the detection number ratio with samples failed library preparation? Even if you failed library preparations, could you still get these results?

L529-564 and 603-605. As mentioned above, I am not convinced that combining method of qPCR and MiFish can reliably estimate the adjusted quantity of DNA in the non-target species of qPCR because MiFish read percentage can vary with other species reads. I recommend authors to delete these sentences and Figure 9-11.

Reviewer #2: (No Response)

7. PLOS authors have the option to publish the peer review history of their article (what does this mean?). If published, this will include your full peer review and any attached files.

Reviewer #1: No

Reviewer #2: **Yes: **Rose Afzali

---

## [Author Response · Author response to Decision Letter 1]

2 Jul 2022

We would like to thank the academic editor and reviewers again for their encouragement and efforts to review the manuscript. We believed we have done our best to address the comments below, including the methodology question raised by reviewer. We much appreciate the reviewers for your thorough and constructive reviews, which we think have greatly improved the manuscript. Our responses are in blue and red (red color marked what we modified in the manuscript), while the original comments from editor and reviewers are in black and italicized, in the "Response_to_Reviewers" file.

Please see details in the "Response_to_Reviewers" file.

Sincerely yours, Shin-ichi Ito

Response to reviewers

Reviewer #1: 

General comments to authors

Authors addressed the reviewer’s comments and revised the manuscript, which got largely improved. Meanwhile, I noticed that this paper does not contain information about negative controls of eDNA samples (i.e., filtering of distilled water or Milli-Q). Because sampling negative controls is important to check possible contamination during field and laboratory processes, authors should add sentences about it. Besides, I do not consider that the combining method of metabarcoding and qPCR results of the target species can provide reliable estimates of eDNA quantity of the non-target species (L238-256, L529-564 and 603-605). Therefore, I recommend authors to delete the related sentences and

figures about this method. I provided detailed comments about these points, as well as minor comments below. 

>Thank you very much for your comments. We are very sorry for our omission of negative control sample here. In the later cruises we collected the negative controls, and the MiFish/qPCR results showed no sign of contamination. We gave the detailed explanation about the problem of negative control below. As for the combining method of metabarcoding and qPCR, we agreed that it is difficult to be confirmed based on the present data, so we deleted the sentences and figures about it (we keep only one figure to illustrate our ideas for the future usage of both methods). And we also addressed other points you raised as showed below.

Detailed comments

L62 The sentence of “The foregoing indicates that~” should be included in the previous paragraph.

> Sorry for our mistake, we have corrected it (L61).

L108 Please add information of “scarce eDNA in open ocean” as “~ in the open ocean, where the concentration of fish eDNAs are expected to be scarce, has been insufficient.”

This will increase a necessity for this study.

> Thank you very much for your advice, so now we wrote it as “~ in the open ocean, where the concentration of fish eDNA is expected to be scarce, has been insufficient.” (L107-108).

L131 Correct to “~were collected from water depths of 5 to 300 m”.

> Sorry for our mistake, we have corrected it (L132).

L131-177 I noticed that this paper does not mention about negative controls of eDNA samples (i.e., filtering of distilled water or Milli-Q). Sampling negative controls is necessary to check possible contamination during field and laboratory processes. Please add sentences about it.

> Thank you very much for pointing out the issue. We developed our water sampling protocols by learning from the previous eDNA studies on marine microorganisms. In those studies, the negative control is not common (like in Stoeck et al. 2010; Behnke et al. 2006). So, our sampling in this study also did not include negative control. However, as you pointed out, we also recognized the importance of sampling negative control in eDNA studies from the later studies. And we performed the negative control sampling in the later cruises (the data from which has not been submitted yet). We performed both qPCR and MiFish methods to the negative controls and we found no sign of contamination. As we used the same sea water sampling protocol in the later cruises, we also speculated that there is also no contamination in the KS-18-5 sampling. 

Stoeck et al. 2010, https://doi.org/10.1111/j.1365-294X.2009.04480.x

Behnke et al. 2006, https://doi.org/10.1128/AEM.72.5.3626-3636.2006

We have added the explanation about the negative control problem in Discussion as follows (L402-413).

“Before discussing about the comparison between the characteristics of qPCR and MiFish methods, we must discuss about data quality of this study. Our seawater sampling protocol was relatively old and it was designed based on previous eDNA studies on the marine microorganisms, in which negative controls were not common [39, 40]. Thus, the KS-18-5 samples did not contain the negative controls. However, the later studies have shown the importance of negative controls to monitor possible contamination during the sampling [41, 42]. Therefore, in the cruises after KS-18-5, negative controls by filtering the Milli-Q water during water sampling for OceanDNA were performed. From the negative controls of the seven cruises after KS-18-15, no target fish DNA have been detected by the qPCR method and no reads have been detected by the MiFish methods (in the form of “library preparation failed as no PCR products obtained”). Although this cannot justify the lack of negative control in KS-18-15, the use of the same seawater sampling method suggested that there could be no contamination in the KS-18-5 sampling.”

L229-232 You should decide this detection threshold of metabarcoding based on the result of negative controls (Yamamoto et al. 2016; Sato et al. 2021). At least, you should show read number of negative controls.

> As we explained above, although we did not have negative control for KS-18-5, in the negative control we collected in later cruises, no read number or qPCR quantity was detected so we speculated there was also no contamination in KS-18-5. Even when contamination was not found, a threshold to avoid false presence is still needed for MiFish (just like in Sato et al. 2021). In our study, we set the threshold as 1% total read in each sample, which is similar but higher than in some previous studies (Li et al. 2019; Hänfling et al. 2016). In our study, the number (sum of all samples) of target OTUs meeting the 1% threshold is 149 and if we halve the threshold (to 0.5%), the number of target OTUs meeting threshold is 158, which only increased slightly. So, we speculated the 1% threshold is not very likely to cause serious change in our results. 

Li et al. 2019, https://doi.org/10.1111/1365-2664.13352

Hänfling et al. 2016, https://doi.org/10.1111/mec.13660

(We also added these two articles into References to support our threshold setting. L200)

L238-256 I understand your calculation for the adjusted DNA quantity of species n of all selected samples. However, I do not consider this calculation reliably estimate DNA quantity of the non-target species of qPCR because MiFish read percentage can vary with read number of other species as shown around 20 copies/μl of qPCR quantity in figure 9. MiFish read percentage can increase or decrease without change of its DNA quantity or read number if read numbers of other species change low or high, respectively. I consider such variability of MiFish read percentage can prevent from estimating reliable DNA quantity of non-target species. Therefore, I recommend authors delete the sentences

about this method and Figure 9-11.

>Thank you very much for your advice. We analyzed and thought it again and as we felt it difficult to confirm the method confidently, so we deleted the sentences about this method and Figure 9, Figure 11. We added a new Fig 9 (modified from original Fig 10a) to illustrate our ideas about future usage of both methods.

L262 Please add information about a depth range or depth layer (bottom or middle?) of net sampling in the main body and caption of Fig.2.

> Thank you very much for your advice. Now we modified the description of net sampling as “The net sampling data used were obtained by a pelagic trawling covered depth of 0-25 m that were performed during a cruise by the R/V Soyo-Maru vessel of the Japan Fisheries Research and Education Agency (net sampling station is shown in Fig 1), on May 11, 2018.”(L245-246), and we also added the information in caption of Fig.2 as “~net sampling (pelagic trawling, at depth of 0-25 m)” (L261). Although the net sampling only covered a narrow depth range, small pelagic fish are known for their diel vertical migration behavior, as showed in (Yasuda et al. 2018; Matsuo et al. 1997; Ohshimo 1996; Stenevik et al. 2007;), so we thought it is better to compare the net sampling data here with OceanDNA sampling data in a wider depth rage. 

Yasuda et al. 2018, https://doi.org/10.3354/meps12636

Matsuo et al. 1997, https://agris.fao.org/agris-search/search.do?recordID=JP1997005975

Ohshimo 1996, https://doi.org/10.2331/fishsci.62.344

Stenevik et al. 2007, https://doi.org/10.2989/AJMS.2007.29.1.12.77

L325 and Fig.4 Checking figure 4, explanation in the caption seems to be incorrect. Pooling-4 and pooling should correspond to (f) and (e), respectively.

>We are very sorry for this mistake. As you said, the pooling-4 and pooling correspond to (f) and (e), respectively. And we have corrected it in the caption of Fig 4 (L308-310).

L488 In the discussion, you should infer possible reasons for preparation failures in MiFish libraries.

> Thank you very much for your advice, now we have added the discussion about the possible reasons for library preparation failures in MiFish as follows (L496-504).

“One critical step in the MiFish library preparation is the first-round PCR that the universal PCR primers amplify the target gene fragments across the target taxa [57]. Therefore, the library preparation failure is likely to happen in the samples with virtually no DNA of target taxa. Indeed, in the negative controls we collected during the later cruises, the library preparation also failed, as mentioned earlier. The lack of target taxa DNA is also possible in seawater samples for the high proportion of microbial DNA and the sparse distribution of fish eDNA [10]. Although MiFish primers were reported to be very effective [58, 59], there are still possible problems like PCR dropout which can lead to amplification failures of existed fish DNA in samples [57]. As shown in our study, practical library preparation of OceanDNA could be challenging in the MiFish method.”

L492-494 How could you still calculate the detection number ratio with samples failed library preparation? Even if you failed library preparations, could you still get these results?

> Sorry for causing the misunderstanding. As you said, if we failed the library preparation in a sample, the MiFish procedure cannot work (like we introduced in the Materials and methods), so target fish detection cannot be found in it. Here we introduced the detection number ratio when included the samples with failed library preparation because we wish to highlight its negative impact to MiFish and call attention from other researchers to deal with the problem of library preparation failure. 

L529-564 and 603-605. As mentioned above, I am not convinced that combining method of qPCR and MiFish can reliably estimate the adjusted quantity of DNA in the non-target species of qPCR because MiFish read percentage can vary with other species reads. I recommend authors to delete these sentences and Figure 9-11.

> Thank you very much for your advice. As mentioned above, we deleted the sentences about the combining method and Figure 9, Figure 11. And we made a little modification of original L 509-528 and used it as the end part of discussion as “Perspectives for future development of OceanDNA analysis methods in marine fish distribution surveys” (L516-559), in which we emphasized the value of both qPCR and MiFish and suggested the further development as follows. Although we agreed the combined method is not reliable at the present, we added a new Fig 9, (modified from original Fig 10a) to illustrate our ideas that combining usage of both methods could be feasible and useful in the future.

“Given the information above, we propose combining the qPCR and MiFish methods in OceanDNA analysis to monitor quantitative distribution of small pelagic fish species with information of fish community structures. The qPCR method provides quantitative estimates and special distribution of target fish species as shown in Figs 5 and 6. On the other hand, using MiFish data of the water samples in which the qPCR method detected the target species (we called them selected samples), fish community coexisting with the target fish can be inferred. For example, we estimated the fish community coexisting with sardine and chub/blue mackerel (Fig 9). This calculation assumed that DNA quantity is proportional to the MiFish read number over the sum of the samples. At this stage, the MiFish data does not have quantitative information. Therefore, the ratios in Fig 9 do not represent relative abundance. To overcome this issue, quantitative sequencing (qSeq) technique, which adds a random sequence tag to the target sequence before PCR in the library preparation to estimate the DNA quantity from read number found in the metabarcoding [61, 62], is required. qSeq has been used in eDNA studies on marine fish, but these studies were performed in environments close to the coast or in the bay [63]. We could not find any previous study on fish with qSeq in the open ocean, although there has been a successful attempt with the microbiome community [64]. Even with qSeq, there is still a risk that the ‘read number-DNA quantity’ relationship varies among different species, which can be examined by qPCR data of several target fish. In addition, the added random sequence tag for qSeq consumed some reads in the sequencing, which may reduce the read number we can obtain from the target fish DNA. In our preliminary experiments (data has not been submitted), adding the sequence tag increased number of non-detected species in MiFish, which suggested the possible difficulty of qSeq in OceanDNA. Thus, we suggest the combination of the qPCR and MiFish methods as a possible way to detect a large number of species that coexist with the target fish. The combined methods will permit a comprehensive understanding of the quantitative distribution of small pelagic fish within fish community in the open waters.”

We also made a little modification in the Abstract and Conclusions and (L44-46, L587-588).

---

## [Decision Letter · Decision Letter 2]

2 Aug 2022

PONE-D-21-36080R2Comparison of species-specific qPCR and metabarcoding methods to detect small pelagic fish distribution from open ocean environmental DNAPLOS ONE

Dear Dr. Ito,

Thank you for submitting your manuscript to PLOS ONE. After careful consideration, we feel that it has merit but does not fully meet PLOS ONE’s publication criteria as it currently stands. Therefore, we invite you to submit a revised version of the manuscript that addresses the points raised during the review process. Please submit your revised manuscript by Sep 16 2022 11:59PM. If you will need more time than this to complete your revisions, please reply to this message or contact the journal office at plosone@plos.org. Please include the following items when submitting your revised manuscript:A rebuttal letter that responds to each point raised by the academic editor and reviewer(s). You should upload this letter as a separate file labeled 'Response to Reviewers'.A marked-up copy of your manuscript that highlights changes made to the original version. You should upload this as a separate file labeled 'Revised Manuscript with Track Changes'.An unmarked version of your revised paper without tracked changes. You should upload this as a separate file labeled 'Manuscript'.If applicable, we recommend that you deposit your laboratory protocols in protocols.io to enhance the reproducibility of your results. Protocols.io assigns your protocol its own identifier (DOI) so that it can be cited independently in the future. For instructions see: https://journals.plos.org/plosone/s/submission-guidelines#loc-laboratory-protocols. Additionally, PLOS ONE offers an option for publishing peer-reviewed Lab Protocol articles, which describe protocols hosted on protocols.io. Read more information on sharing protocols at https://plos.org/protocols?utm_medium=editorial-email&utm_source=authorletters&utm_campaign=protocols.

We look forward to receiving your revised manuscript.

Kind regards,

Arga Chandrashekar Anil, Ph. D., D. Agr.,

Academic Editor

PLOS ONE

Journal Requirements:

Reviewers' comments:

Reviewer's Responses to Questions

**Comments to the Author**

1. If the authors have adequately addressed your comments raised in a previous round of review and you feel that this manuscript is now acceptable for publication, you may indicate that here to bypass the “Comments to the Author” section, enter your conflict of interest statement in the “Confidential to Editor” section, and submit your "Accept" recommendation.

Reviewer #1: (No Response)

2. Is the manuscript technically sound, and do the data support the conclusions?

Reviewer #1: Yes

3. Has the statistical analysis been performed appropriately and rigorously? 

Reviewer #1: Yes

4. Have the authors made all data underlying the findings in their manuscript fully available?

Reviewer #1: Yes

5. Is the manuscript presented in an intelligible fashion and written in standard English?

Reviewer #1: Yes

6. Review Comments to the Author

Reviewer #1: Authors addressed the reviewer’s comments and improved the manuscript. I just provided minor comments below.

Minor comments to authors

L62 Correct to “OceanDNA has the potential to be a valuable fish survey method” .

L402-413 I understand your excuses regarding no negative controls in this study. In these sentences, both KS-18-5 and KS-18-15 appeared, and confusing. The KS-18-15 was mistakenly mentioned? Or such a cruise was also present?

L526 Correct to “three metabarcoding primers”.

L547-548 The qSeq is different from the qMiseq used in Ushio et al. (2018), so the reference of [63] is incorrect. The latter method uses internal standard DNAs to create sample-specific regression lines between the numbers of sequence reads and DNA copies for estimating original DNA copy numbers.

7. PLOS authors have the option to publish the peer review history of their article (what does this mean?). If published, this will include your full peer review and any attached files.

Reviewer #1: No

---

## [Author Response · Author response to Decision Letter 2]

12 Aug 2022

We would like to thank the academic editor and reviewers again for their encouragement, and efforts and patience to review the manuscript. We revised our manuscript based on the new comments. Again, we really appreciate the reviewers for your thorough and constructive reviews, which we think have greatly improved the manuscript. 

Please see the detail response in the Our responses are in the Response file.

Thank you.

---

## [Editor Report · Decision Letter 3]

15 Aug 2022

Comparison of species-specific qPCR and metabarcoding methods to detect small pelagic fish distribution from open ocean environmental DNA

PONE-D-21-36080R3

Dear Dr. Ito,

We’re pleased to inform you that your manuscript has been judged scientifically suitable for publication and will be formally accepted for publication once it meets all outstanding technical requirements.

Kind regards,

Arga Chandrashekar Anil, Ph. D., D. Agr.,

Academic Editor

PLOS ONE
---

## [Editor Report · Acceptance letter]

17 Aug 2022

PONE-D-21-36080R3 

Comparison of species-specific qPCR and metabarcoding methods to detect small pelagic fish distribution from open ocean environmental DNA 

Dear Dr. Ito:

I'm pleased to inform you that your manuscript has been deemed suitable for publication in PLOS ONE. Congratulations! Your manuscript is now with our production department. 

Kind regards, 

on behalf of

Professor Arga Chandrashekar Anil 

Academic Editor

PLOS ONE